# A complex between IF2 and NusA suggests early coupling of transcription-translation

Mikhail Metelev ® & Magnus Johansson ® ✉

The main function of translation initiation factors is to assist ribosomes in selecting the correct reading frame on an mRNA. This process has been extensively studied with the help of reconstituted in vitro systems, but the dynamics in living cells have not been characterized. In this study, we performed single-molecule tracking of the bacterial initiation factors IF2, IF3, as well as the initiator fMet-tRNA$^{fMet}$ directly in growing *Escherichia coli* cells. Our results reveal the kinetics of factor association with the ribosome and, among other things, highlight the respective antagonistic roles of IF2 and IF3 in the process. Importantly, our comparisons of in vivo binding kinetics of two naturally occurring isoforms of IF2 reveal that the longer IF2α isoform directly interacts with the transcriptional factor NusA, a finding further corroborated by pull-down and cross-linking experiments. Our results suggest that this interaction may promote formation of a coupled transcription-translation complex early in the translation cycle, motivating further structural studies to validate the mechanism. We further show that cells with compromised binding between IF2α and NusA display slow adaptation to new growth conditions.

Rapid and accurate initiation of mRNA translation is crucial for any living cell as it defines the reading frame of the mRNA and the production output for each protein. Canonical translation initiation can be divided into a number of steps, i.e., the small ribosomal subunit (30S) finding the correct initiation site on mRNA, correct positioning on the start codon where initiator tRNA (fMet-tRNA$^{fMet}$) base-pairs with mRNA, docking of the large ribosomal subunit (50S), and finally maturation of the complex (70S) into an elongation competent state (Fig. 1a). Base-pairing between the 3'-end of 16S rRNA, known as anti-Shine-Dalgarno, and a corresponding mRNA motif, called the Shine-Dalgarno sequence, were thought to provide the primary mechanism with which ribosomes locate the initiation sites[1]. However, recent studies show that other mRNA features collectively play a larger role in this process[2], among which the thermodynamic stability of secondary structures surrounding the start codon might be the dominant factor[3,4]. To define the reading frame by correct positioning of fMet-tRNA$^{fMet}$ on the AUG start codon, ribosomes further rely on the coordinated action of three initiation factors (IFs), IF1, IF2, and IF3[5].

Translation initiation factor 1 (IF1), encoded by the *infA* gene, is the smallest of the IFs. It consists of 71 aa (8.2 kDa) and binds to the A site of the 30S subunit. IF1 provides additional binding points for IF2 and IF3, and has been shown to stabilize their interaction and enhance their activities[6]. The gene *infB* encodes IF2, a large GTPase which recruits fMet-tRNA$^{fMet}$ via its C-terminal domain and facilitates 50S subunit joining to the 30S initiation complex. In *E. coli*, the *infB* gene contains three alternative in-frame translation initiation codons, leading to the production of three IF2 isoforms. The two smaller isoforms, IF2β (79.9 kDa) and IF2γ (78.8 kDa), differ by only seven amino acids in the N-terminus, while the significantly larger IF2α isoform (97.3 kDa) includes an additional 157-amino-acid-long Domain I[7,8]. Both the large (IF2α) and the smaller isoforms (IF2β and IF2γ) are required for optimal growth[8], and they are present in similar concentrations[9], suggesting that the isoforms might have overlapping but not identical functions. The functional difference between the three isoforms is poorly understood, though reports are indicating that the IF2α isoform might play a role in maintaining genome integrity[10,11] and in protein folding[12]. Finally, the gene *infC* encodes the two-domain

Department of Cell & Molecular Biology, Uppsala University, Uppsala, Sweden. ✉e-mail: m.johansson@icm.uu.se

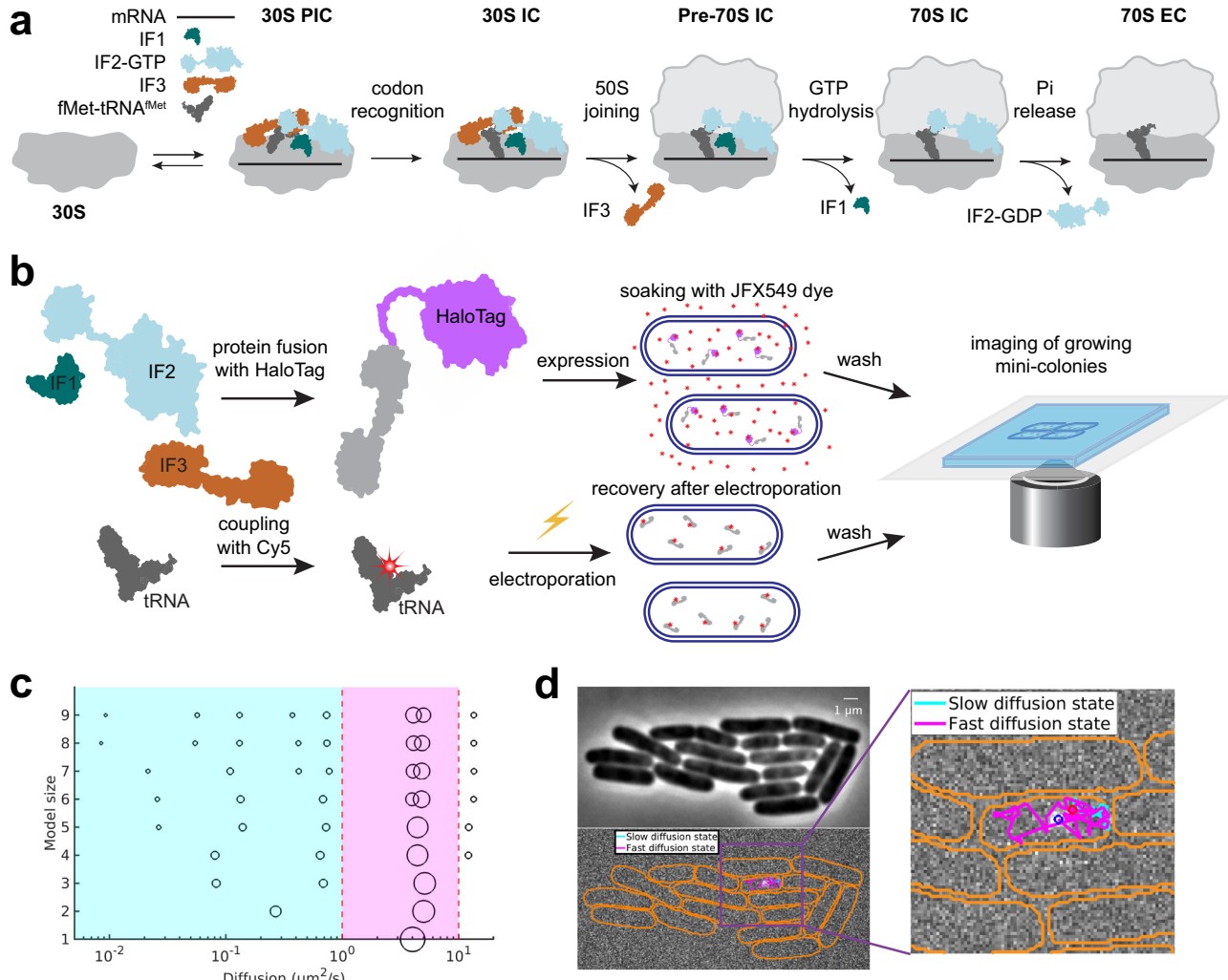

**Fig. 1 | Experimental design for single-molecule tracking. a** Schematic of the main steps and key intermediates in canonical translation initiation. **b** Two approaches for fluorescence labeling were used: (upper) IFs fused to the HaloTag protein were expressed at low concentration from a plasmid with subsequent coupling to the JFX549 fluorophore in vivo, and (lower) labeling of fMet-tRNA$^{fMet}$ in vitro with the Cy5 fluorophore, followed by the delivery of labeled molecules via electroporation. **c** HMM fitting of single-molecule trajectories of HaloTag-IF3 using models with 1–9 states (Supplementary Data 3). Each model is arranged along the y axis, with discrete diffusion states represented by circles positioned along the x-axis according to their diffusion coefficient. The area of each circle represents the relative occupancy of different diffusional states. Red dashed lines indicate thresholds at 1 μm²/s and 8 μm²/s, used to distinguish ribosome-bound states (cyan area) from free factors (magenta area) and cleavage products. **d** Phase contrast images and fluorescence time-lapse movies were acquired for mini-colonies of cells that were recovered after labeling. These images were used for cell segmentation and for constructing single-molecule diffusion trajectories within the corresponding cells. Cell outlines are shown in orange. Beginning and the end of the trajectories are shown with red and blue circles, respectively. The figure shows a diffusion trajectory of HaloTag-IF3 (see also Movie S4), color coded with respect to binding state (cyan and magenta). Source data are provided as a Source Data file.

protein IF3 which prevents premature 50S subunit association and ensures fidelity of fMet-tRNA$^{fMet}$ selection by monitoring the codon:anticodon interaction with mRNA and destabilizing interactions with incorrect elongator tRNAs[13–17].

Single-molecule in vitro studies and stopped-flow techniques in bulk have previously been employed to study the timing and the order of factor association during assembly of the canonical 30S initiation complex[18,19]. These studies, however, yielded contradictory results. One study suggests that complex formation may proceed via multiple pathways[19], while the other advocates for the existence of a single kinetically favored route[18]. That is, IF3 and IF2 are the first to bind the 30S. Thereafter, the arrival of IF1 leads to the formation of a stable complex which recruits fMet-tRNA$^{fMet}$. Upon correct base pairing between the fMet-tRNA$^{fMet}$ anticodon and the start codon, the 50S subunit joins. This triggers hydrolysis of IF2 bound GTP[20], with subsequent dissociation of all IFs[21] and the final maturation of the 70S initiation complex into an elongation competent state[20,22,23].

Decades of studies on IFs have shown how they mediate the key steps of translation initiation, and novel techniques continue to reveal their structural dynamics and the overall underlying complexity of the process[23–25]. However, the majority of these studies rely on reconstituted in vitro systems with short model mRNAs, and it is, still, difficult to predict how IFs and ribosomes behave in the complex environment of the cell. For example, there is growing evidence that the IFs, especially IF2, might be involved in other processes such as ribosome maturation, protein folding, stress adaptation, DNA repair, and transcription-translation coupling[10,12,26–28].

In this study, we developed an approach for in vivo single-molecule tracking of IF2 and IF3 directly in live *E. coli* cells. By measuring the respective steady-state fraction and dwell time of each factor bound to ribosomes, we show that on average, fMet-tRNA$^{fMet}$ is the last to arrive at the 30S. IF2 samples the ribosomes multiple times during initiation, and binds only transiently without prompt arrival of the fMet-tRNA$^{fMet}$. Interestingly, we observed significant differences in

the diffusion behavior of the IF2α and IF2γ isoforms, suggesting that the longer IF2α isoform has an additional major interaction partner besides the ribosomes in the cells. With additional experiments, we demonstrate that Domain I of IF2α, which is absent in the other two isoforms, is responsible for frequent and distinct binding to the transcription machinery via the C-terminal domain of the transcription factor NusA. Hence, our results suggest that a physical connection between the transcription and translation machineries might be established already during translation initiation.

## Results

### Labeling of IFs and fMet-tRNA$^{fMet}$

To fluorescence label the IFs for live-cell single-molecule tracking, we explored if the addition of a HaloTag compromises IF activity in supporting cell growth (Supplementary Note 1). For IF2, we included both the largest isoform, IF2α, and the smallest, IF2γ. We observed that expression of only the IF2α isoform can support growth rate similar to the wild-type (WT) strain (Supplementary Fig. 1b), while cells expressing only the IF2γ isoform grow slower even at higher expression levels (Supplementary Fig. 1c). For neither of the IF2 isoforms, the addition of HaloTag at the N-terminus altered their functionality (Supplementary Fig. 1b, c). The addition of HaloTag has no or only a slight negative effect on IF3 function (Supplementary Fig. 1d), whereas HaloTag addition noticeably affects the functionality of IF1 (Supplementary Fig. 1a). In agreement with these results, it has been previously shown that a dCasRx-IF3 fusion was functional in targeting the ribosomes to specific mRNAs, while a dCasRx-IF1 fusion showed only limited capacity in ribosome targeting[29].

We next prepared modified plasmids for single-molecule tracking in a WT genetic background. That is, by introducing a weak SD sequence, we achieved fluorescence labeling of only a few molecules per cell, in a close to unperturbed background. For single-molecule tracking experiments, we employed an approach that we have previously used to track fluorescently labeled 30S and 50S ribosomal subunits (Fig. 1b)[30]. Briefly, cells expressing one of the fusion proteins were grown to exponential phase, and then mixed with a chloroalkane-activated JFX549 dye which forms a covalent bond with HaloTag[31]. The cells were then extensively washed to remove unbound dye, and finally sparsely placed on agarose pads and grown to mini-colonies at 37 °C. This procedure results in an even distribution of labeled molecules in growing and dividing cells. For each fusion protein, more than 1000 individual cells in mini-colonies were imaged using stroboscopic laser illumination (3 ms) at 5 ms camera exposure times. The microscopy movies were analyzed using a previously developed semi-automatic pipeline that segments cells in the mini-colonies based on phase-contrast images, detects fluorescent dots in each frame, and builds trajectories of diffusing fluorescent molecules from frame to frame (Fig. 1b). In the microscopy experiments we observed on average less than 10 fluorescent molecules per cell. Since HaloTag labeling is very efficient ($\geq 80\%$)[32] and there are thousands of non-labeled IFs per cell[33], we conclude that the total number of each IF does not change significantly due to the labeling approach with leakage-level expression from a plasmid, and hence, that the binding kinetics observed should represent that of an unperturbed *E. coli* cell.

In addition to IF tracking, we also performed tracking of fMet-[Cy5] tRNA$^{fMet}$ as described previously[34]. To deliver Cy5-labeled fMet-tRNA$^{fMet}$ into bacteria we employed our previously established procedure for electroporation (Fig. 1b)[34]. After electroporation, we followed the same protocol for imaging as for the HaloTag-labeled molecules, but with SYTOX Blue in the agarose pad. SYTOX Blue is a dye that selectively stains dead cells, and was included so that cells that did not recover after electroporation could be discarded in the analysis. Imaging was performed on growing mini-colonies with more than 4 cells per colony as an additional precaution to ensure that cells had recovered from electroporation[35]. The efficiency of fMet-tRNA$^{fMet}$ delivery during

electroporation varied greatly between mini-colonies, but after dilution by growth, the cells contained only a small fraction of labeled molecules relative to the endogenous levels. Hence, as with the HaloTag-labeled factors, the addition of Cy5-labeled fMet-tRNA$^{fMet}$ should not significantly affect the total concentration of fMet-tRNA$^{fMet}$ and therefore not perturb the overall translation initiation kinetics.

### HMM analysis of single-molecule trajectories

Concentrations of IFs and fMet-tRNA$^{fMet}$ in *E. coli* cells are linked to the growth rate and the number of ribosomes. Fast-growing bacteria contain roughly equimolar amounts of each factor at ~25 % of the concentration of total ribosomes[9,36]. The IFs and fMet-tRNA$^{fMet}$ are at the same time present in significant excess (2x-3x) over the fraction of ribosomal subunits that are currently involved in translation initiation, i.e., the 10% of total ribosomes that are not actively translating[30]. Hence, we expected to detect IFs in two main diffusion states: a larger fraction of free molecules diffusing quickly and a smaller ribosome-bound fraction diffusing slower. For all the factors fluorescently labeled using either of the two approaches, we observed the expected diffusion behavior, i.e., random fast diffusion throughout the complete cell cytoplasm, interrupted by slow diffusion events, consistent with usage in translation initiation (Supplementary Movie 1-5). In addition to tracking of IFs and fMet-tRNA$^{fMet}$, we also included HaloTag protein itself and our previously developed HaloTag labeled 30S[30] in this study for comparison (Supplementary Movie 6). Distributions of diffusion step lengths for each individual factor support the observation that the tracked factors occupy at least two diffusion states, one similar to that of the ribosomes and one with faster diffusion (Supplementary Fig. 2).

To quantify the steady-state distribution of factors in the different diffusion states and, particularly, the frequency with which the factors transition between these states, we used a Hidden-Markov-Modelling (HMM) approach that we have previously applied to study binding kinetics of tRNAs, SRP, as well as ribosomal subunits[30,34,35]. In the algorithm, all trajectories are fitted using global maximum-likelihood estimation, to a model with a pre-defined number of hidden diffusion states, where diffusion coefficients and transition frequencies between the diffusion states within trajectories are used as fitting parameters[37]. The algorithm also accounts for motion blur, localization uncertainty, and potential missing positions in trajectories. The high brightness and stability of the JFX549 dye allowed us to track IFs for up to hundreds of frames, with average trajectory lengths around 30 frames (Supplementary Fig. 3). Long trajectory length allows us to capture not only the distribution of molecules in different diffusion states but also capture transitions between states within single trajectories (Fig. 1d and Supplementary Fig. 4). It should also be noted that dwell times in different states are calculated based on the HMM fitted frequencies of transitions between different states, and not on duration of complete binding events. Hence, with the assumption that the system of study is in steady-state, it is possible to estimate longer binding events than the recorded trajectory lengths.

In biological systems, variations in local environments and complexity of interactions typically result in distributions of diffusion rates, rather than well-defined discrete diffusion states. In our previous single-molecule-tracking study of ribosomal subunits, we found highly heterogeneous diffusion encompassing a wide range – from relatively fast-diffusing single 30S and 50S, to slowly diffusing polysomes, which may also be tethered to other macromolecular complexes, such as translocons in the cell membrane[30]. This heterogeneity is apparent also in histograms of diffusion step lengths, which are more accurately modeled by multiple diffusion states rather than only one (Supplementary Fig. 5). In the HMM analysis, this complexity can be modeled using multiple discrete states, which are subsequently grouped into biologically meaningful categories through an additional coarse-graining step. Importantly, coarse-graining of larger models also

helps to capture non-Markovian transition behavior, including non-exponential time distributions in diffusion states (see discussion in[30]).

The diffusion trajectories of all tracked molecules in the current study were HMM-fitted to models with 1–9 diffusion states (Supplementary Data 1-7). Similar to results in our previous studies on tracking of tRNAs, SRP, as well as ribosomal subunits, we found that, according to Akaike's information criterion (AIC), larger models with more states fit the data better (Supplementary Fig. 6). With an increasing number of HMM-fitted states we found that 30S subunits display a wide range of diffusion states spanning from ~0.01 $\mu m^2$/s (polysomes) to ~1 $\mu m^2$/s (free diffusion), dependent on the translational status of the ribosomes (as discussed previously in detail[30]) (Supplementary Fig. 7a, Supplementary Data 6). Although a fraction of the unfused Halotag proteins showed slow diffusion as well, the occupancy of those states was much lower (<4%) and with no apparent wide distribution of diffusion coefficients consistent with ribosome binding (Supplementary Fig. 7b, Supplementary Data 7). We hence conclude that HaloTag itself does bind to something slowly diffusing in the cell, potentially giving rise to a small background in our data of HaloTag-labeled components.

With an increasing number of HMM-fitted states, we observed a common cluster of diffusion states for each of the IFs and for fMet-tRNA[fMet] similar to that of the ribosomes, i.e., between 0.01 and 1 $\mu m^2$/s (Supplementary Fig. 7c-g and Supplementary Data 1-5). In addition, we found a second cluster of diffusion states with significantly faster diffusion (>1.9 $\mu m^2$/s) which were factor-specific and showed a negative correlation with the molecular size (Supplementary Data 1-5). We also observed that for each individual dataset for HaloTag-IF2α, HaloTag-IF2γ, HaloTag-IF3, as well as for HaloTag-labeled 30S ribosomes, in fittings with ≥5 states, there is an additional state (<9 %) with even higher diffusion coefficient >8 $\mu m^2$/s, even for the ribosomes, which likely corresponds to a cleavage product of HaloTag fusions which releases free labeled HaloTag (Supplementary Fig. 7a, 7d-f, and Supplementary Data 8-10). Such state could not be distinguished for HaloTag-IF1 due to the small size of IF1 which likely causes poor separation between HaloTag-IF1 and HaloTag (Supplementary Data 11). Hence, with an assumption that the main function of the IFs in the cell is to mediate translation initiation, we assign the first cluster of slow diffusion states (0-1 $\mu m^2$/s) to factors associated with ribosomes, the second cluster of diffusion states (1−8 $\mu m^2$/s) corresponding to freely diffusing factors, and the diffusion states >8 $\mu m^2$/s as proteolysis products for HaloTag-IF2α, HaloTag-IF2γ, and HaloTag-IF3.

Since our aim was to investigate ribosome binding kinetics of the factors, we, thus, coarse-grained the multi-state models for fMet-tRNA[fMet], HaloTag-IF1 and HaloTag down to a 2-state model (using thresholds of 1 $\mu m^2$/s to differentiate ribosome-bound from free factors), and for HaloTag-IF2α, HaloTag-IF2γ, HaloTag-IF3 to a 3-state model (using thresholds of 1 $\mu m^2$/s and 8 $\mu m^2$/s to differentiate ribosome-bound from free factors and cleavage products) (Supplementary Fig. 8).

Based on the described HMM-analysis with subsequent coarse-graining of diffusion states, we first find that HaloTag-IF1 showed a significantly lower proportion in the presumably ribosome-bound state (~7 %) than other labeled IFs, and only marginally higher than what is observed for HaloTag alone (~4 % in the slow diffusion state) (Supplementary Data 4, 7, 11, 13). Considering also that HaloTag-IF1 could not completely compensate for the lack of IF1 in our growth assays (Supplementary Fig. 1a), we concluded that HaloTag-IF1 is not sufficiently functional, and we have, therefore, omitted it from further analysis.

### Ribosome binding kinetics of IFs and fMet-tRNA[fMet]

For clarity and simplicity, we discuss below coarse-grained results for occupancies and dwell times from HMM analysis of tracking data for all labeled components using a model size of 5 diffusion states, where we observe a robust separation into three clusters: ribosome-bound, free,

and cleavage products (Fig. 2a, b). While absolute values vary slightly across different model sizes, our conclusions remain consistent for all models ≥5 states (Supplementary Fig. 9), with occupancies being very robust between different HMM model sizes and dwell-times showing higher variability.

From the coarse-grained results, we find that the steady-state occupancy of IFs in the presumed ribosome-bound state varies between 21% and 57%, whereas the steady-state fraction of fMet-tRNA[fMet] bound to ribosomes is only 5% (in line with previous results[34,38,39]) (Fig. 2a, Supplementary Data 8-10, 12). Furthermore, based on the fitted frequency of transitions between the slow and the fast diffusion states, we calculated the average dwell-times in the respective states (Fig. 2b, Supplementary Data 8-10, 12). We note, however, that the observed bindings might not necessarily correspond only to initiation events, but can be a combination of productive and unproductive bindings. In an attempt to evaluate if a factor binds to the ribosome more than one time per translation cycle, we calculated an average factor cycling time as the sum of the dwell times in the slow and the fast diffusion states (Fig. 2b). The theoretical average time required for one translation cycle for a ribosome can be derived from the average length of all proteins, the average elongation rate, as well as the average time to recycle the ribosomal subunits between translation events. With this calculation (see Supplementary Note 2), we estimate that the average translation cycle, from the perspective of the ribosomal subunits, is 10–17 s. Further, the average productive cycling time for the ribosomes and factors respectively should be proportional to their relative concentrations. While the tRNA/ribosome ratio can be measured with high precision, around one tRNA[fMet] per four ribosomes under fast growth conditions[36], precise measurement of absolute protein concentrations in the cell is much more challenging and depends on the extraction protocol and quantification method. The metadata on protein abundance levels suggests that the concentrations of IF2 and IF3 are ~20% and ~37%, respectively, of that of the ribosomes[40]. With these assumptions, we can thus conclude that a cycle time of <2.0 s for IF2, <2.5 s for fMet-tRNA[fMet], and <3.7 s for IF3 would indicate more than one ribosome binding event per translation cycle.

We observe that fMet-tRNA[fMet] shows the lowest occupancy in the slow-diffusion state, ~5% (4−9 times lower than for the IFs), as well as the shortest dwell time in the slow-diffusion state, 0.18 s +/- 0.03 s (Fig. 2, Supplementary Data 12). Since fMet-tRNA[fMet] dissociates from the ribosome after one elongation cycle (Fig. 1a), and hence leaves last, these numbers strongly suggest that fMet-tRNA[fMet] also arrives at the ribosome last, in agreement with biochemical studies[18]. Further, the average cycle time for fMet-tRNA[fMet] is ~3.3 s (Fig. 2b) which indicates that the majority of the detected binding events are productive.

We see that labeled IF2 and IF3 are efficiently and dynamically used by the translation machinery (Fig. 2, Supplementary Data 8-10). For IF3 we observe a steady-state fraction of bound molecules at 21% +/- 3% (Fig. 2a, Supplementary Data 10). For IF2, we observe that both isoforms show higher occupancy in the slow-diffusion state compared to IF3, but interestingly different from each other (47% +/- 4% and 57% +/- 2% for IF2γ and IF2α, respectively) (Fig. 2a, Supplementary Data 8, 9). Despite the fact that both IF2 isoforms show significantly higher occupancy in the slow-diffusion state in comparison with fMet-tRNA[fMet], they display only moderately longer dwell times in the slow-diffusion state (0.26 s +/- 0.05 s and 0.34 s +/- 0.06 s, for IF2α and IF2γ, respectively, compared to 0.18 s +/- 0.03 s for fMet-tRNA[fMet]) (Fig. 2b). The average cycle time for IF2α and IF2γ is 0.44 s +/- 0.07 s and 0.72 s +/- 0.07 s, respectively (Fig. 2b). These numbers are significantly lower than the calculated threshold of 2.0 s above. Taken together, these results strongly indicate that the complex between IF2 and the ribosome is labile, and that IF2 binds to ribosomes more frequently than once per translation event. For IF3 we observe the longest dwell time in the slow-diffusion state, 0.38 ms, with a longer average

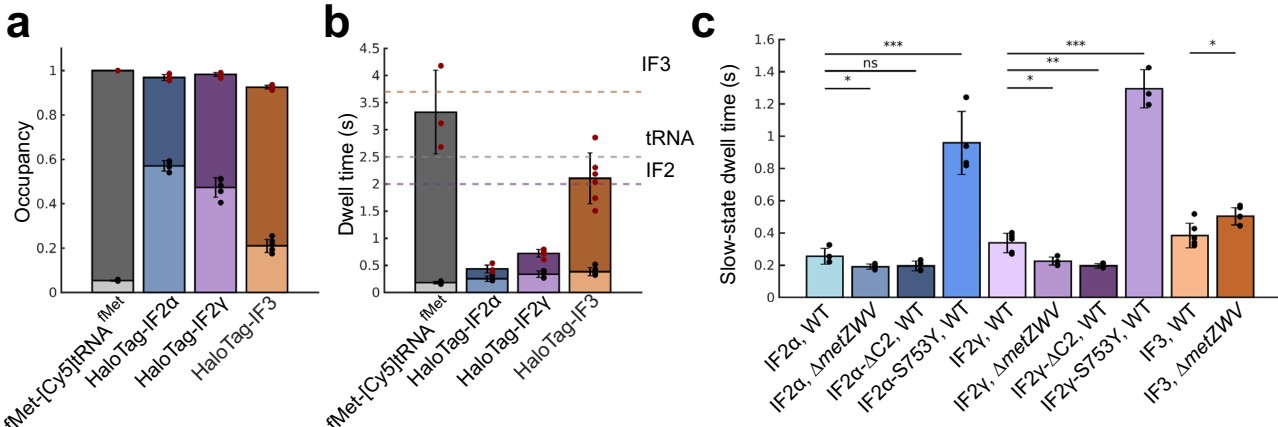

**Fig. 2 | Binding kinetics of IFs and fMet-tRNA^fMet from coarse-grained HMM-results. a, b** Estimated occupancies (**a**) and dwell-times (**b**) of labeled factors in the "bound" (bottom bar) and the "free" (top bar) diffusion states. Dashed lines in (**b**) are thresholds for estimates of the expected calculated total time for a single initiation event. The violet, grey and orange dashed lines indicate the thresholds at 2.0 s, 2.5, and 3.7 seconds, for IF2, fMet-tRNA^fMet, and IF3, respectively. The data show results from coarse-grained 5-state HMM models, with averages calculated from independent experiments. Error bars represent standard deviations between these independent experiments. The combined number of cells and trajectory steps were as follows: fMet-tRNA^fMet – 2039 cells and 53,849 steps in 3 independent experiments; HaloTag-IF2α – 1592 cells and 134,697 steps in 4 independent experiments; HaloTag-IF2γ – 1622 cells and 127,210 steps in 5 independent experiments; and HaloTag-IF3 – 2491 cells and 172,989 steps in 6 independent experiments. Results for the 5-state models are provided in Supplementary Data 8-10, 12. Results for all model sizes for combined data are provided in Supplementary Data 1-3, 5. The results from coarse-grained HMM models of size 5–9 are shown in Supplementary Fig. S9. **c** Estimated dwell-times of HaloTag labeled IF3 as well as IF2 isoforms and mutants in the "bound" diffusion state. In the *ΔmetZWV* strain, three of four genes for tRNA^fMet are deleted, resulting in lower in vivo concentration of fMet-tRNA^fMet. The data show results from coarse-grained 5-state HMM models, with averages calculated from independent experiments. Error bars represent standard deviations between these independent experiments. Results for the 5-state models are provided in Supplementary Data 8-10, 18-24. The combined

number of cells and trajectory steps were as follows: HaloTag-IF2α – 1,592 cells and 134,697 steps in 4 independent experiments; HaloTag-IF2α in the *ΔmetZWV* strain – 1399 cells and 110,116 steps in 4 independent experiments; HaloTag-IF2α-ΔC2 – 1429 cells and 118,712 steps in 4 independent experiments; HaloTag-IF2α-S753Y – 1825 cells and 131,921 steps in 4 independent experiments; HaloTag-IF2γ – 1622 cells and 127,210 steps in 5 independent experiments; HaloTag-IF2γ in the *ΔmetZWV* strain – 1609 cells and 127,467 steps in 4 independent experiments; HaloTag-IF2γ-ΔC2 – 1606 cells and 140,172 steps in 4 independent experiments; HaloTag-IF2γ-S753Y – 1727 cells and 139,119 steps in 3 independent experiments; HaloTag-IF3 – 2,491 cells and 172,989 steps in 6 independent experiments; and HaloTag-IF3 in the *ΔmetZWV* – 2919 cells and 246,146 steps in 6 independent experiments. Results for the all model sizes for combined data are provided in Supplementary Data 1-3, 25-31. Statistical significance between groups was assessed using a two-sided unpaired *t*-test. *P*-values are indicated in the figure as follows: $P < 0.05$ (*), $P < 0.01$ (**), $P < 0.001$ (***), and not significant (ns) otherwise. The *p*-values comparing IF2α in the WT strain with IF2α in the *ΔmetZWV* strain, IF2α-ΔC2, and IF2α-S753Y are 0.0489, 0.0867, and 0.0004, respectively. The *p*-values comparing IF2γ in the WT strain with IF2γ in the *ΔmetZWV* strain, IF2γ-ΔC2, and IF2γ-S753Y are 0.0102, 0.0026, and $4.5 \times 10^{-6}$, respectively. The *p*-values comparing IF3 in the WT strain with IF3 in the *ΔmetZWV* strain is 0.0102. The results from coarse-grained HMM models of size 5–9 are shown in Supplementary Fig. S12. Source data are provided as a Source Data file.

cycle time at 2.1 s (Fig. 2b), closer, but still lower than the theoretical cycle time of 3.7 s. Hence, our results suggest that IF2 and IF3 have different binding behavior to the ribosome. That is, IF2 ribosome binding is more dynamic than IF3's and probably requires more binding attempts per translation initiation event.

As a control for the HaloTag-labeled components, we tracked a K110L mutant IF3, deficient in 30S interaction[41], as well as an IF2 variant lacking Domain I and II, which has been shown to possess dramatically lower affinity to ribosomes[42]. For both IF2 and IF3 mutants, we observed a large decrease in slow-state occupancy (12% +/- 5% and 5 % +/- 1%, respectively, versus 57% +/- 2% (IF2α) and 21% +/- 3% for the non-mutated versions), showing that the proposed ribosome bindings events of the non-mutated HaloTag-labeled factors are specific (Supplementary Fig. 10, Supplementary Data 8-10, 13-17).

In the analysis presented above, the HMM results from larger models were coarse-grained into three primary states: 30S-associated factors, free factors, and cleavage products/tracking artifacts. In our previous study on ribosome tracking[30], we distinguished between ribosomal subunits bound to mRNAs and those diffusing freely by applying a threshold of 0.25 μm²/s. In that study, we used longer camera exposure times (30 and 60 ms, relative to 5 ms in the present study), which provided higher resolution for slow-diffusing molecules and facilitated better coverage of transitions between the long-lived biological states (bound-state dwell times up to 60 s). In the present study, in HMM models with ≥4 states for all labeled IFs, we identify two

distinct ribosome-associated state sub-clusters with diffusion coefficients of approximately 0.1 μm²/s and 0.5–0.7 μm²/s (Supplementary Fig. S7, Supplementary Data 1-4, 8-11). These likely correspond to initiation factors associated with 30S already bound to a transcript (IF-30S-mRNA) and 30S not yet engaged with an mRNA (IF-30S), respectively, mirroring observations from our previous ribosome tracking studies. Notably, the occupancies of these sub-clusters remained consistent across different HMM model sizes, with the following distributions for IF-30S-mRNA and IF-30S states: 38%/20% for IF2α, 33%/14% for IF2γ, and 11%/10% for IF3. This suggests that both IF2 isoforms are more enriched in the 30S-mRNA state compared to IF3, in agreement with IF3's earlier arrival and departure – occurring before or immediately after 50S association – whereas IF2 remains associated with the 70S complex during GTP hydrolysis and Pi release (Fig. 1a).

However, HMM-estimated transitions between the IF-30S-mRNA/IF-30S sub-states and the free factor state were model-dependent, in the sense that the estimated steady-state fluxes of molecules between the different states were not consistent between coarse-grained models based on different larger multi-state models (Supplementary Fig. S11). A likely explanation is that while the 5 ms camera exposure times used in this study enables the detection of shorter binding events and the tracking of smaller, faster-diffusing molecules, it does not resolve transitions between slow-diffusion states as effectively as with longer exposure times. Hence, although the HMM analysis of IF diffusion trajectories clearly suggest a distinction between

IF-30S-mRNA and IF-30S sub-states, we cannot achieve robust kinetics data at this level of detail. Thus, our reported IF bound-state dwell times correspond to the average binding of the respective IF to 30S subunits in both these sub-states.

## fMet-tRNA[fMet] affects 30S-binding by IF2 and IF3 differently

To further investigate the dynamics of IF binding, we decided to disturb the system, first by delaying the arrival of fMet-tRNA[fMet]. The *E. coli* genome contains four genes encoding fMet-tRNA[fMet], three of which are located in the *metZWV* operon. Cell growth can, however, be supported by the fourth gene, *metY*, alone, albeit at a slower rate due to the lower concentration of the initiator tRNA[43]. We investigated the binding kinetics of HaloTag versions of IF3 and both IF2 isoforms in *E. coli ΔmetZWV*.

For IF3, we observed an approximately 30% increase in the dwell time in the ribosome-bound state in cells with a lower level of fMet-tRNA[fMet], suggesting that the lower fMet-tRNA[fMet] concentration delays the formation of the initiation complex and extends the binding of IF3 (Fig. 2c, Supplementary Data 10, 24). This is in perfect agreement with results from a reconstituted system suggesting that fMet-tRNA[fMet] facilitates the dissociation of IF3 from 30S[16]. On the contrary, for IF2γ, we find that the decrease of fMet-tRNA[fMet] concentration caused a 33% decrease in the average dwell time in the ribosome-bound state, suggesting that the complex between IF2γ and the ribosome is not stable without fMet-tRNA[fMet] (Fig. 2c, Supplementary Data 9, 19). For the IF2α isoform, we also observed a similar trend in the slow-state dwell-time between WT and the *ΔmetZWV* mutant for the 5-state model (Fig. 2c, Supplementary Data 8, 18). However, the difference and its significance were not consistent across all different HMM models (Supplementary Fig. S12).

To further explore how binding of fMet-tRNA[fMet] affects the stability of the ribosome-IF2 complex, we performed tracking of the two IF2 isoforms, now with deleted Domain VI-2 which is crucial for the interaction with fMet-tRNA[fMet] [44]. Similar to the experiments in the *ΔmetZWV* strain, we observed a 42 % decrease in the slow-diffusion state dwell time for HaloTag-IF2γ-ΔC2 and no significant decrease for HaloTag-IF2α-ΔC2 in comparison with the non-mutated isoforms (Fig. 2c, Supplementary Data 8, 9, 20, 21). These results support our finding that fMet-tRNA[fMet] binding is essential to stabilize IF2 on the 30S ribosome.

Our results from IF2γ tracking are in agreement with ensemble kinetics studies[45] and single-molecule FRET experiments, which suggest that IF2's interaction with GTP and fMet-tRNA[fMet] promotes activation of IF2 and stabilizes its high-affinity conformation on the 30S subunit[24]. Hence, without fast arrival of fMet-tRNA[fMet], IF2γ is not able to adopt this active conformation and dissociates from the complex. Quite remarkably, a single mutation, S753Y, causes IF2 to preferentially adopt the active conformation independent of fMet-tRNA[fMet] [24,45]. Single-molecule tracking of both IF2 isoforms containing the S753Y substitution also reveals dramatically altered ribosome binding kinetics, with dwell times in the slow-diffusion state of 0.96 +/- 0.20 s and 1.29 s +/- 0.12 for IF2α and IF2γ, respectively, compared to 0.26 +/- 0.05 s and 0.34 +/- 0.06 s for the WT factors (Fig. 2c, Supplementary Data 8, 9, 22, 23). We speculate that for these mutants, the dwell time in the bound state likely corresponds to the average time for complete, productive initiation events, since shorter unproductive events should here be minimal. We cannot rule out, however, that the mutation also affects the departure of IF2 during 70S complex maturation.

To summarize, we find that WT IF2 does not form a stable complex with the ribosome without prompt arrival of fMet-tRNA[fMet]. Interestingly, the S753Y mutation, which dramatically extends the ribosome-bound time of IF2, has previously been shown to increase the rate of erroneous translation initiation with elongator tRNA, and is also associated with a reduced fitness[46]. Hence, taken together, these results suggest that stable binding of IF2 to the ribosome serves as an important checkpoint – all involved components must be present and in correct conformation for IF2 to stably bind and allow the process to continue. The extension of the ribosome-bound dwell-time of IF3 at lower fMet-tRNA[fMet] concentration, on the other hand, suggests a completely different role of this factor in the regulation of translation initiation fidelity. That is, IF3 maintains 30S in an initiation-competent state, prohibiting 50S joining, until the correct aminoacylated tRNA has arrived. Finally, we observed a noticeable difference in diffusion behavior between the two isoforms of IF2: IF2α showed a larger steady-state fraction in the slow-diffusion state compared to IF2γ, but with a shorter slow-diffusion-state dwell time. The two isoforms also responded differently to the functional mutations. These kinetic discrepancies between the isoforms motivated us to further investigate possible functional differences between the two IF2 isoforms.

## The N-terminal Domain I of IF2α has an alternative binding partner in the cell

Compared to IF2γ, the IF2α isoform contains an additional 50 amino-acid long N-terminal compact sub-domain N1, and a flexible linker connecting it to the C-terminal domains[47,48] (Fig. 3a). Such structural domain has been predicted to be evolutionarily conserved, and is likely present in most bacteria[47]. In *E. coli*, two such structural sub-domains are present. The second copy, the N2 sub-domain, is located right before the G-domain and likely plays a role in 50S-subunit association[49]. The function of the N1 sub-domain, only present in the IF2α isoform, has, however, remained obscure[49]. Interestingly, the production of several isoforms of IF2 with varying N-terminus is not limited to *E. coli* but has been shown for several other bacteria, including species of *Pseudomonas*, *Streptococcus*, *Sarcina*, and *Bacillus*[50,51], indicating that diverse bacteria produce two or more different isoforms of IF2 with specialized functions.

To investigate the function of IF2 Domain I, we created a Domain-I-HaloTag fusion and tracked its diffusion in *E. coli*. HMM analysis of single-molecule trajectories of Domain-I-HaloTag revealed that it displays transient (~30 ms) and frequent transitions to a slow-diffusion state, with a steady-state fraction in this slow-diffusion state of 14% +/- 2% (Fig. 3b, c, Supplementary Data 32, 33). Previous studies have suggested that IF2 Domain I does not noticeably interact with ribosomes[42]. Thus, the observed slow-diffusion state might represent binding to an alternative partner with slow diffusion in the cell environment. The spatial distribution of IF2 Domain I in the slow-diffusion state appears to be similar to that of nucleoid-associated proteins (Fig. 3d), such as RNA Polymerase (RNAP) (Fig. 3e, Supplementary Data 34)[52]. The slow diffusion state distribution is, therefore, also anti-correlated with the overall spatial distribution of the ribosomes which are mainly excluded from the nucleoid (Fig. 3f)[52]. Hence, our single-molecule tracking experiments suggest an alternative binding partner for IF2α, mediated by Domain I, spatially overlapping with the nucleoid. Although Domain I of IF2α shows strong co-localization with the nucleoid, the spatial distributions of the slow-diffusion states of IF2α and IF2γ are not clearly distinct, likely due to their shared primary function – binding to ribosomes – which masks more subtle differences between the two isoforms (Supplementary Fig. 13).

## The N1 domain of IF2 binds to NusA

Besides the well-characterized role in canonical translation initiation, IF2 has been shown to play a role in the late stages of ribosome maturation[12,53]. Involvement in ribosome maturation is further supported by the discovery of an antibiotic, lamotrigine, which binds IF2 and compromises ribosome biogenesis[54,55]. On the other hand, IF2 has also been identified to participate in an interaction network that includes transcription factor NusA and RNA polymerase[56]. However, to the best of our knowledge, it has not been investigated if the link between IF2 and the transcription machinery is direct. Interestingly and related, the IF2 gene, *infB*, is located in an operon between the

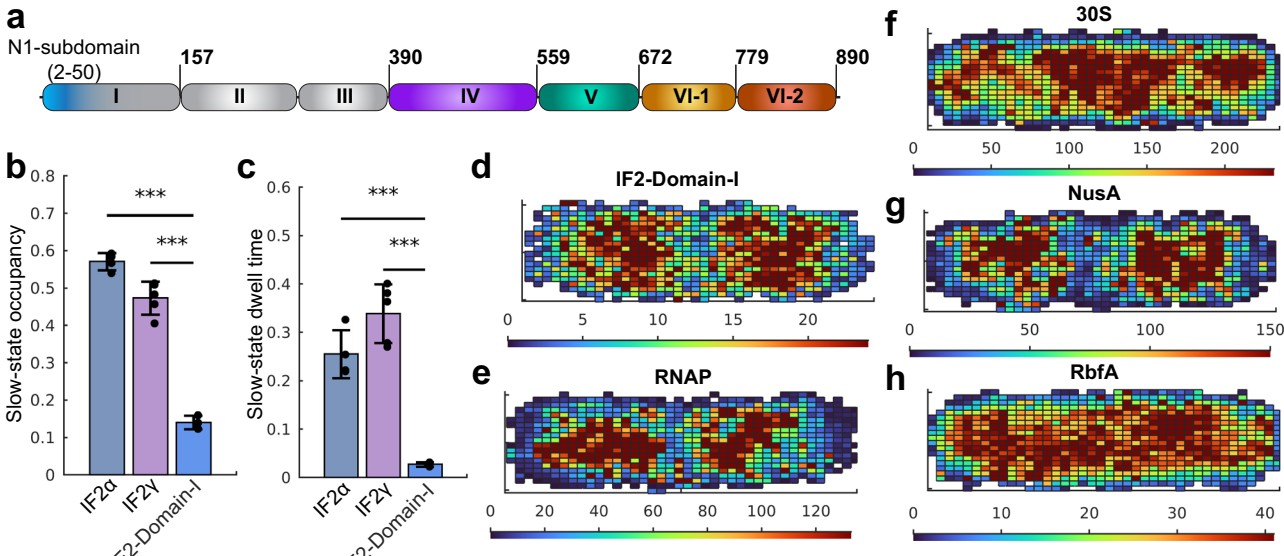

**Fig. 3 | The IF2 Domain I has an alternative binding partner other than the ribosome. a** Schematic representation of domain organization of IF2α. Domain I is absent in the IF2β and IF2γ isoforms of IF2. **b, c** Estimated occupancies (**b**) and dwell-times (**c**) of HaloTag labeled IF2α, IF2γ, and IF2 Domain I in the slow diffusion state (<1 μm²/s). The data show results from coarse-grained 5-state HMM models, with averages calculated from independent experiments. Error bars represent standard deviations between these independent experiments. The combined number of cells and trajectory steps were as follows: HaloTag-IF2α – 1592 cells and 134,697 steps in 4 independent experiments; HaloTag-IF2γ – 1622 cells and 127,210 steps in 5 independent experiments; and IF2-Domain-I-HaloTag – 3423 cells and 280,151 steps in 3 independent experiments. Results for the 5-state models are provided in Supplementary Data 8, 9, and 33. Results for all model sizes for combined data are provided in Supplementary Data 1, 2, and 32. Statistical significance between groups was assessed using a two-sided unpaired *t*-test. *P*-values are indicated in the figure as follows: $P < 0.001$ (***). The *p*-values comparing HaloTag-IF2α and HaloTag-IF2γ with IF2-Domain-I-HaloTag are $1.4 \times 10^{-6}$ and $1.9 \times 10^{-5}$, respectively, for slow-state steady-state occupancies (**b**), and $5.7 \times 10^{-4}$ and $1.3 \times 10^{-4}$ for slow-state dwell times (**c**). **d–h** Spatial distribution in the slow diffusion state of HaloTag-IF2-Domain-I (**d**) from 5-state model, RpoC-HaloTag (**e**) from 1-state model, h6-MS2CP-HaloTag labeled 30S (**f**) from 1-state model, NusA-HaloTag (**g**) from 5-state model, and RbfA-HaloTag (**h**) from 5-state model (with coarse graining threshold at 1.2 μm²/s). To limit variability in nucleoid number and positioning due to cell cycle stage[52], we applied size-based filtering to exclude newly divided cells, characterized by a single nucleoid at mid-cell or two poorly separated lobes, as well as larger cells with more than two nucleoid lobes, in line with previous studies[52]. The combined number of cells and trajectory steps after cell sorting were as follows: IF2-Domain-I-HaloTag – 1091 cells and 91,267 trajectory steps in 3 independent experiments; RNAP (rpoC-HaloTag) – 500 cells and 48,208 trajectory steps in 3 independent experiments, 30S (h6-MS2CP-HaloTag) – 842 cells and 102,165 trajectory steps in 3 independent experiments; NusA-HaloTag – 762 cells and 71,823 trajectory steps in 3 independent experiments; RbfA-HaloTag – 686 cells and 39,683 trajectory steps in 7 independent experiments. Calculated occupancy in the slow state for IF2-Domain-I-HaloTag, NusA-HaloTag, and RbfA-HaloTag, is 11%, 78% and 50%, respectively. All states were plotted for RNAP and 30S. Source data are provided as a Source Data file.

genes encoding the multifunctional transcription factor NusA and the ribosome biogenesis factor RbfA. RbfA is required for efficient 16S rRNA processing and acts as a checkpoint to ensure complete maturation of the 30S subunits that are ready to enter the pool of initiating ribosomal subunits[28,57,58].

While the intracellular diffusion of RNA polymerase and transcription factors has been extensively studied[52,59,60], the diffusion of partially assembled ribosomal particles during ribosome biogenesis has, to our knowledge, not been investigated. Maturation of the 30S subunits takes ~60 seconds and starts before transcription of rRNA is completed[61]. While 16S rRNA is still transcribed, such incomplete 30S are likely to be tethered to the chromosome through RNA polymerase, at least during the earlier phase before transcription termination (i.e. the first 30 s[61]). If Domain I of IF2α is responsible for factor binding to incomplete 30S, this could explain its spatial distribution in the cell. As an initial naïve test to find the additional binding partner of IF2α, we decided to compare the diffusion behavior of RbfA and NusA with Domain I of IF2. To this end, we performed single-molecule tracking of RbfA-HaloTag and NusA-HaloTag fusions. Based on the results from the HMM analysis, we observe two main diffusive clusters for NusA-HaloTag – a slow-diffusion state with a diffusion coefficient of 0.1 μm²/s and a nucleoid-like spatial profile (Fig. 3g), and a fast-diffusion state at 4.5 μm²/s distributed homogeneously in the cell (Supplementary Fig. 14a, Supplementary Data 35, 36). RbfA-HaloTag, on the other hand, showed a very different diffusion behavior, with two main diffusion

states at ~1 μm²/s and ~9 μm²/s, both distributed homogeneously in the cell (Fig. 3h, Supplementary Fig. 14b, Supplementary Data 37, 38), suggesting that RbfA acts at the stage when rRNA is already released from the transcription complex, in agreement with reports showing that it is involved in the late stage of 30S maturation[28,62,63]. Since IF2 has been suggested to be involved only in the final checkpoint of ribosome maturation[53], we conclude that the slow-diffusion state of Domain I and nucleoid-like spatial distribution (Fig. 3d) is not compatible with the hypothesis that Domain I interacts with maturing ribosomes.

As mentioned above, an interaction between IF2 and NusA has been detected previously[56]. In this large-scale study, the authors used pull-down and mass spectrometry to identify protein-protein interactions in *E. coli*[56]. We fused a polyhistidine tag to IF2α, IF2γ, and isolated Domain I of IF2, and performed affinity purification of these proteins to investigate if NusA co-elutes with the tagged proteins. The experiment was performed in a strain where the HaloTag gene was fused to NusA on the chromosome to facilitate its detection. Affinity purification of tagged IF2α confirmed that it co-purifies with NusA-HaloTag (Fig. 4a). No such interaction was detected for IF2γ, strongly indicating that Domain I is necessary for the interaction. We also conclude that Domain I is sufficient for the interaction since large amounts of NusA-HaloTag were co-purified during affinity chromatography of Domain I alone (Fig. 4a). No interaction was detected between HaloTag alone and either of the IF2 isoforms or Domain I (Supplementary Fig. 15).

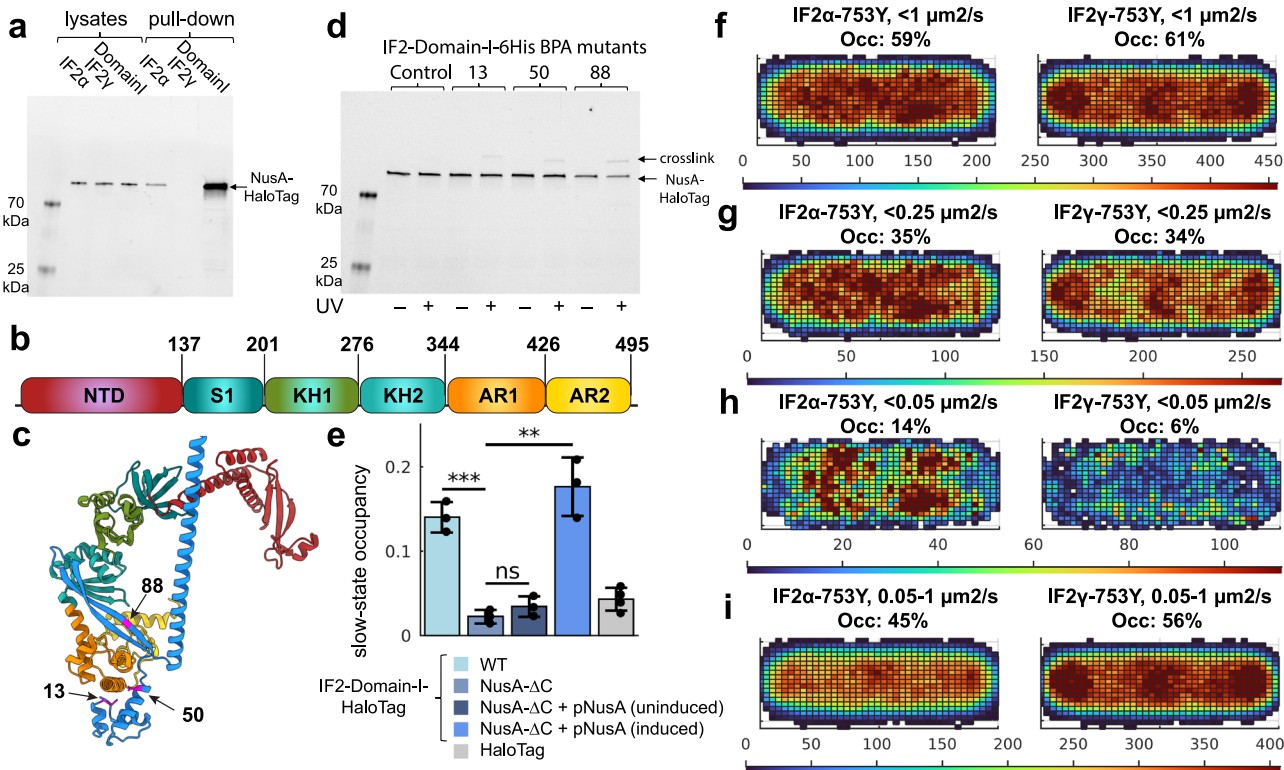

**Fig. 4 | IF2 Domain I interacts with the C-terminal domain of NusA. a** SDS-PAGE analysis of cell lysates and affinity purification fractions for the presence of NusA-HaloTag. 6His-IF2α, 6His-IF2γ, and IF2-Domain-I-6His were overexpressed in an *E. coli* strain in which the gene for HaloTag was inserted at the C-terminus of the chromosomal locus for NusA. Lysates and samples affinity purified on Co-NTA agarose (i.e. "pull-down") were loaded on SDS-PAGE gel and stained using the JFX549 dye to label the NusA-HaloTag fusion. Each experiment was independently repeated twice. **b** Schematic representation of domain organization of NusA. **c** Alphafold prediction of the complex between IF2 Domain I (blue) and NusA (colored according to the domain organization shown on panel (**b**). Positions 13, 50, and 88, selected for insertion of the unnatural BPA cross-linking amino acid are shown in magenta. **d** Affinity purification of overexpressed IF2 Domain I (WT control and BPA containing mutants) from cell cultures untreated or treated with UV for cross-linking. An *E. coli* strain with chromosomal *nusA-HaloTag* was used for overexpression. After affinity purification on Co-NTA agarose, samples were loaded on an SDS-PAGE gel and stained using JFX549 dye to label NusA-HaloTag fusion and cross-linked products. **e** Estimated occupancies in the slow diffusion state of HaloTag tracked in WT *E. coli* strain and HaloTag-IF2-Domain I in WT, NusA-*ΔC* mutant strain, and NusA-*ΔC* supplemented with the pNusA plasmid in which the *nusA* gene is regulated by the inducible P_BAD promoter. The data show results from coarse-grained 5-state HMM models, with averages calculated from independent experiments. Error bars represent standard deviations between these independent experiments. The combined number of cells and trajectory steps were as follows: IF2-Domain-I-HaloTag in WT strain – 3,423 cells and 280,151 steps in 3 independent experiments; IF2-Domain-I-HaloTag in NusA-*ΔC* strain – 2,274 cells and 134,556 steps in 3 independent experiments; IF2-Domain-I-HaloTag in NusA-*ΔC*

strain with pNusA plasmid without inducer – 3254 cells and 181,628 steps in 3 independent experiments; IF2-Domain-I-HaloTag in NusA-*ΔC* strain with pNusA plasmid with inducer – 3309 cells and 197,951 steps in 3 independent experiments; and HaloTag in WT strain – 2630 cells and 109,253 steps in 4 independent experiments. Results for the 5-state models are provided in Supplementary Data 33, 39-41. Results for all model sizes for combined data are provided in Supplementary Data 32, 42-44. Statistical significance between groups was assessed using a two-sided unpaired *t*-test. *P*-values are indicated in the figure as follows: $P < 0.01$ (**), $P < 0.001$ (***), and not significant (ns) otherwise. The *p*-value comparing IF2-Domain-I-HaloTag in the WT strain and in the NusA-*ΔC* strain is $4.9 \times 10^{-4}$. The *p*-values comparing IF2-Domain-I-HaloTag in the NusA-*ΔC* strain with the same strain carrying the pNusA plasmid, with or without inducer, are 0.2240 and 0.0017, respectively. **f–i** Spatial distribution of the slow diffusion state of HaloTag-IF2α-S753Y and HaloTag-IF2γ-S753Y, extracted from 5-state HMM models using coarse-graining with diffusion threshold at $1\,\mu m^2/s$ (**f**), $0.25\,\mu m^2/s$ (**g**), $0.05\,\mu m^2/s$ (**h**), and slow diffusional state in the range of $0.05 - 1\,\mu m^2/s$ (**i**). To maximize the amount of data for spatial distribution analysis, tracking was performed allowing detection of up to two dots per cell, rather than a single dot per cell used in the previous experiments. Size-based filtering was applied to exclude newly divided cells, characterized by a single nucleoid at mid-cell or two poorly separated lobes, as well as larger cells with more than two nucleoid lobes[52]. The combined number of cells and trajectory steps after cell sorting were as follows: HaloTag-IF2α−753Y – 2221 cells and 337,000 trajectory steps in 3 independent experiments; HaloTag-IF2γ−753Y – 2354 cells and 328,151 trajectory steps in 3 independent experiments. Occ - occupancy in the slow state after coarse graining with different thresholds. Source data are provided as a Source Data file.

The pull-down experiment shows that Domain I of IF2 indeed interacts with NusA. However, it does not exclude the possibility that the interaction is indirect. To further evaluate the interaction, we first used AlphaFold 3[64] to predict a structure of the possible complex between Domain I of IF2 and NusA. The AlphaFold prediction suggests that the interactions are predominantly mediated by the N1 structural domain (1-50 aa) and the following sequence forming two β-strands (57-90 aa) on IF2, and the C-terminal part including domains KH2 and AR1 (Fig. 4b, c) on NusA. We then used the orthogonal photo-crosslinkable amino acid benzoyl-phenylalanine (pBpa) to investigate if Domain I of IF2 directly interacts with NusA in *E. coli* cells. For this, we

constructed different mutants of IF2 Domain I in which the pBPA amino acid was incorporated in several positions (amino acids 13, 50, and 88), predicted from the AlphaFold structure to be in close proximity with NusA (Fig. 4c). The IF2 Domain I mutants were then expressed in *E. coli* harboring a chromosomal NusA-HaloTag, and crosslinks were induced by UV illumination. For each selected Domain I position, we observed cross-linking to NusA-HaloTag, providing evidence that Domain I of IF2 interacts directly with NusA in the cell (Fig. 4d).

The AlphaFold model predicted that the interaction with IF2 is mediated by the domains KH2 and AR1 of NusA (Fig. 4c). The

C-terminal fragment of NusA that includes the AR1 and AR2 domains is not essential for cell viability and can be deleted[65]. This allowed us to validate our finding by investigating the diffusion behavior of the IF2 Domain I in the absence of its potential corresponding interaction domain on NusA. Hence, we tracked IF2 Domain I in cells with the C-terminal fragment of NusA deleted from the chromosome, and the results were striking. The short nucleoid-associated bindings that were detected in the WT strain virtually disappeared in the strain expressing truncated NusA (2% +/- 1% slow-state occupancy vs 14% +/- 2% in the WT strain, Fig. 4e). On the other hand, if full-length NusA was expressed from an inducible plasmid in cells with the chromosomally mutated NusA gene, the diffusion pattern was restored (Fig. 4e). Hence, we conclude that IF2 and NusA interact with each other via their N-terminal and C-terminal domain, respectively.

Interestingly, we observe similar diffusion coefficients for IF2 Domain I and NusA in the free diffusion state (4-5 µm²/s, Supplementary Data 33, 35), suggesting that IF2α and NusA might establish their interaction while freely diffusing. To explore this further, we investigated whether NusA can interact with IF2 and/or ribosomes independently of RNAP by performing single-molecule tracking of a NusA variant lacking its N-terminal domain (ΔNTD-NusA-HaloTag), the primary region responsible for binding RNAP (Fig. 4b). The experiments were conducted in both WT cells and in a strain lacking the IF2α isoform, generated by a chromosomal deletion of the gene segment encoding IF2 Domain I (ΔIF2-Domain-I), allowing us to specifically evaluate the dependency of NusA interactions on IF2α. HMM analysis revealed that in the ΔIF2-Domain-I strain, ΔNTD-NusA-HaloTag shows only background-level occupancy (5% +/- 1%) in the slow diffusion state (<1 µm²/s), indicating that it does not associate with macromolecular complexes such as RNAP and ribosomes (Supplementary Fig. S16, Supplementary Data 45-47). In contrast, in the WT strain, with intact IF2 Domain I, ΔNTD-NusA-HaloTag showed a modest increase in slow-state occupancy (8% ± 1%), likely reflecting interactions with ribosomes via IF2α (Supplementary Fig. S16, Supplementary Data 48-50). Furthermore, analysis of larger HMM models (Supplementary Fig. S17, Supplementary Data 46, 47, 49, 50) shows that ΔNTD-NusA-HaloTag displays a larger fraction in diffusion states similar to that of IF2α (in the range of 1−3 µm²/s) in the WT background compared to the ΔIF2-Domain-I strain. Hence, we conclude that to some degree, NusA can interact with IF2α, via Domain I, even in solution and on the ribosome without simultaneous RNAP binding, albeit to a limited extent.

## Isoform-specific dynamics of IF2 in the bacterial nucleoid
Frequent but short-lived (~30 ms) interactions between Domain I of IF2α and NusA bound to transcribing RNAP likely contribute to the different diffusion behaviors observed for the two IF2 isoforms in our microscopy experiments. We observed a higher fraction of IF2α in the slow-diffusion state, consistent with a mixed population of molecules bound either to ribosomes or transiently interacting with NusA-RNAP complexes. On average, IF2α displays shorter binding durations than IF2γ, which likely reflects its engagement in both stable ribosome-associated interactions and brief samplings of the transcription machinery. We speculate that these transient interactions may facilitate translation on nascent mRNAs indirectly, for example, by increasing the local concentration of IF2 within the nucleoid and thereby promoting rapid translation initiation. It is also possible that such interactions can promote the formation of a coupled complex involving IF2α, the ribosome, NusA, and RNAP. Directly capturing such complexes with our microscopy techniques, however, is inherently challenging – the interactions between IF2α and either the ribosome or the NusA-RNAP complex are rather transient (tens to hundreds of milliseconds); and even if IF2 mediates a ribosome-RNAP connection via NusA, such events are expected to be rare, as only the leading ribosome on a nascent mRNA could physically interact with RNAP, while subsequent translation initiation events are unlikely to be influenced. Nevertheless, in an attempt to specifically investigate the existence of such Nusa-IF2α mediated ribosome-RNAP connection, we utilized an IF2-S753Y mutant, which has previously been shown to possess stabilized ribosome binding[24,45]. Imaging of HaloTag-IF2α-S753Y was carried out at a slower acquisition rate (30 ms per frame) in a ΔIF2-Domain-I strain. The slower frame rate reduces the sensitivity to short-lived interactions with NusA and improves the resolution of slow-diffusion states, while using the ΔIF2-Domain-I background eliminates competition from endogenous IF2α isoforms. For comparison, we tracked HaloTag-IF2γ-S753Y under identical conditions, allowing us to isolate isoform-specific effects.

Analysis of spatial distributions shows that, with progressively lower diffusion thresholds for coarse-graining (1 µm²/s, 0.25 µm²/s, and 0.05 µm²/s), the slower-diffusing HaloTag-IF2α-S753Y becomes increasingly enriched in the nucleoid region (Fig. 4f-h, Supplementary Data 51-53). Notably, a substantial fraction of molecules (~14%) exhibit very slow diffusion (<0.05 µm²/s), preferentially localizing within the nucleoid and with dwell-time estimates of ≥1 s (Fig. 4h, Supplementary Data 53). In contrast, HaloTag-IF2γ-S753Y shows a different spatial distribution: molecules with diffusion coefficients <1 µm²/s and <0.25 µm²/s are preferentially enriched at the cell poles and mid-cell, where most translation occurs (Fig. 4f–h, Supplementary Data 54-56). Furthermore, the fraction of HaloTag-IF2γ-S753Y molecules exhibiting very slow diffusion (<0.05 µm²/s) is considerably lower (~6%), with these binding events enriched, to some extent within the nucleoid, but mainly along the cell periphery (Fig. 4h, Supplementary Data 54-56). Finally, we find that the slow-diffusing HaloTag-IF2α-S753Y population, excluding the slowest state from 0 µm²/s to 0.05 µm²/s, displays a more homogeneous distribution, with mild enrichment at the cell poles and mid-cell (Fig. 4i). This suggests that it is specifically the slowest-diffusing states that are nucleoid-associated, while the remaining population behaves similarly to the HaloTag-IF2γ-S753Y. Taken together, our microscopy experiments, hence, suggest that the interaction between the IF2α isoform and NusA likely promotes the formation of a long-lived coupled complex between the ribosome and RNAP, mediated by their respective associations with IF2α and NusA. However, confirming the existence and nature of such a complex will require further structural studies.

## Effect of IF2 Domain I deletion on cell growth
Previous studies have shown that the presence of all IF2 isoforms is required for fast *E. coli* growth[8]. Our growth experiments show that plasmid-expressed IF2α can support WT-level growth in a ΔinfB strain, whereas the strain expressing only IF2γ shows slower growth (Supplementary Fig. 1). To further study the effect of the different IF2 isoforms, we created a strain with a chromosomal deletion of the gene fragment corresponding to the IF2 Domain I (ΔIF2-Domain-I). Hence, in this strain, IF2α is absent, but both of the shorter isoforms, IF2β and IF2γ, can be expressed. The growth defect associated with the lack of IF2 Domain I was moderate in rich LB (18% slower) and minimal media M9 supplemented with 0.4% glucose as a carbon source (15% slower), but more pronounced in M9 supplemented with 0.4% succinate (41% slower) (Fig. 5a). Interestingly, we observed that the mutant strain needs longer time to adapt to the new environment after a change of growth media, i.e., a prolonged lag phase during adaptation to the new environment (Fig. 5b, c). Both WT and the ΔIF2-Domain-I strain that were grown overnight in either M9-glucose or M9-succinate ("adapted" strains) were able to quickly resume growth when diluted in fresh media of the same type (Fig. 5b, c, yellow and blue curves). When the strains were grown in rich LB media ("unadapted" strains), a clear extension of the lag phase was shown when the cultures were diluted into M9-glucose (Fig. 5b) or M9-succinate (Fig. 5c, orange and purple curves). However, the lag phase after the media change was significantly longer for the ΔIF2-Domain-I strain than for the WT strain, particularly in M9-succinate (10 hours). The lag phase is the most

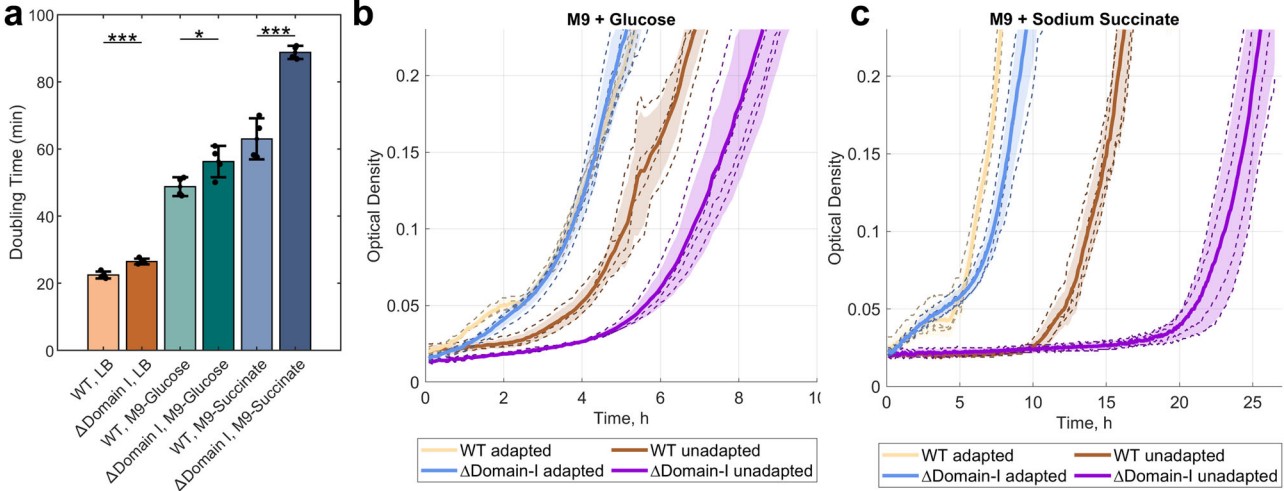

**Fig. 5 | Growth defect of *E. coli* ΔIF2-Domain-I strain. a** Doubling time of WT and *ΔIF2*-Domain-I strains in LB and M9 media supplemented with glucose or succinate as a carbon source. Bars and error bars represent mean and standard deviation, respectively, calculated from 4 independent biological replicates, where each individually measured doubling time is shown as a circle. Statistical significance between groups was assessed using a two-sided unpaired *t*-test. *P*-values are indicated in the figure as follows: $P < 0.05$ (*), $P < 0.001$ (***). The *p*-values comparing WT and *ΔIF2*-Domain-I strains grown in LB, M9 supplemented with glucose, and M9 supplemented with succinate are 0.0009, 0.0335, and 0.0002, respectively. **b, c** Growth curves of WT and *ΔIF2*-Domain-I in M9 media supplemented with glucose (**b**) or succinate (**c**) after growing for 24 h either in LB ("unadapted" strains) or in the M9 media ("adapted" strains). All strains were diluted in fresh media to the same starting optical density. Mean growth curves for each strain and condition are shown as solid lines, individual biological replicas are shown as dashed lines, and calculated standard deviations are shown as a shadow. Source data are provided as a Source Data file.

poorly understood stage of bacterial growth but is critical for bacterial fitness in a changing environment[66]. During the lag phase, the bacteria undergo dynamic transcriptional and translation changes[66–68]. Hence, our results suggest that the interaction between IF2 and NusA is important particularly in transcription and translation rewiring during adaptation to a new environment.

## Discussion

The main macromolecular complexes in bacteria, such as the translation and transcription machineries, are highly robust and can perform their function in simplified reconstituted systems. Over the past decades, research exploiting such reconstituted systems has allowed us to dissect the key functional properties of the main components and auxiliary factors. In the cell environment, however, these processes are tightly coordinated, connected, and highly organized to achieve optimal speed, fidelity, usage of the available resources, and adaptability to a changing environment. A comprehensive understanding of biology on the molecular level, hence, requires the development of methods that can capture this additional layer of complexity directly in living cells. Single-molecule fluorescence microscopy in live cells is a unique approach that can complement the extensive knowledge collected from reconstituted in vitro systems, and expand our understanding of how key macromolecular complexes function in the crowded and interconnected cell environment.

We have previously applied our single-molecule-tracking approach to study the kinetics of translation by directly following the movement of initiator and elongator tRNAs, as well as ribosomal subunits[30,34,38,39]. In the present study, we were able to capture diffusion dynamics of IF2 and IF3 in live *E. coli* cells. Analysis of ribosome binding kinetics of IFs and fMet-tRNA$^{fMet}$ reveal that initiator tRNA is the last to bind to the ribosome during translation initiation in agreement with in vitro studies in bulk. We furthermore observe that IF2, the main factor facilitating the binding of fMet-tRNA$^{fMet}$, does not form a stable complex with the ribosome prior to arrival of fMet-tRNA$^{fMet}$. Such transient complex formation likely helps to tune the accuracy of translation initiation, given that a single point mutation, which renders IF2 to bind ribosomes with higher affinity, is associated with a decrease in accuracy of initiation and compromised fitness[45,46]. We note that

both IF2α and IF2γ isoforms display such behavior, but our conclusions are based primarily on results with IF2γ, due to the fact that IF2α binds not only the ribosome but also the transcription machinery. Moreover, IF2 and fMet-tRNA$^{fMet}$ show different diffusion coefficients in the fast, non-ribosome-bound, diffusion state (2-3 μm²/s and 10 μm²/s, respectively), strongly indicating that they do not form a stable complex off the ribosome in the cell. Hence, our results suggest that IF2 does not deliver the initiator fMet-tRNA$^{fMet}$ to the ribosome, clarifying a controversy based on results in reconstituted systems[19,69]. IF3, on the other hand, displays more stable binding to the ribosomes than IF2, in line with its main role of preventing premature association with 50S.

Further examination of the IF2 isoforms revealed apparent differences in diffusion properties and binding kinetics, consistent with an additional binding partner for the larger IF2α isoform. Our work establishes that the N-terminal Domain I, present only in the IF2α isoform, is responsible for binding to the C-terminal region of NusA.

NusA is a multifunctional, multi-domain protein that regulates transcription elongation, pausing, and termination[70–73]. It contains a conserved N-terminal domain (NTD) that anchors it to RNAP, three RNA-binding domains (S1, KH1, and KH2), and two less conserved acidic repeat domains (AR1 and AR2) at the C-terminus. Off the RNAP, NusA adopts an autoinhibitory conformation due to internal interactions between its AR2 and KH1 domains. This conformation is relieved upon binding to RNAP, where the NTD engages the β-flap tip helix and AR2 binds the α-subunit C-terminal domain (α-CTD)[74,75]. ChIP-chip data suggest that NusA is a general transcription modulator, engaging early after departure of RNAP from the promoter region, and is uniformly associated with RNAP molecules on most transcription units[70]. NusA alone promotes transcription pausing and termination but it also serves as a binding platform for multiple regulatory factors modulating its activity. NusA plays a central role in the assembly of the anti-termination complex on rRNA operons, where it cooperates with NusB, NusE, NusG, ribosomal protein S4, and the inositol monophosphatase SuhB[72]. This complex is structurally distinct from the conventional paused elongation complex[71,76] and it enables RNAP to transcribe long, structured operons by suppressing intrinsic termination signals.

Beyond its established functions in regulating transcription elongation and termination, NusA has also been implicated in transcription-translation coupling (TTC). Cross-linking mass-spectrometry, structural in vitro studies, as well as cellular cryo-electron tomography have shown that NusA, together with the universally conserved transcription factor NusG, can bind both the RNAP and the ribosome, providing a physical link for transcription-translation coupling[77–80]. While such physical coupling has been observed in evolutionarily distant organisms such as *E. coli* and *Mycoplasma*, and hence may represent a common feature among bacteria, its mechanistic implementation differs between the two species, highlighting the evolutionary diversity of TTC systems.

Our findings suggest that the interaction between NusA and IF2 Domain I may represent an additional layer of coordination between transcription and translation. Our in vivo data show that IF2 via its Domain I exhibit frequent but short-lived bindings to NusA associated with the RNAP. AlphaFold3 modeling predicts that Domain I of IF2 interacts with the KH2 and AR1 domains of NusA in a conformation resembling that of the paused RNAP complex. This interaction appears incompatible with the antitermination conformation of NusA, which adopts a significantly different structural arrangement and includes a steric clash with ribosomal protein S4. This suggests that IF2 is unlikely to engage NusA when NusA is part of the rRNA antitermination complex. Further structural and biochemical studies will be needed to shed light on the architecture of IF2-NusA interactions[81]. Our in vivo results also show that IF2 and NusA can interact in solution, although the interaction appears limited, likely due to the predominance of the autoinhibited NusA conformation in the absence of RNAP.

The majority of research on transcription-translation coupling has been focused on how translation elongation stimulates transcription[73,81]. However, several studies have shown that the transcription machinery can assist in translation initiation and thus stimulate an early establishment of coupling, with the prominent example of RfaH, a specialized paralog of NusG[63,82]. However, RfaH is present in *E. coli* cells at very low concentration, and regulates expression of only a small subset of operons. In the present work, we reveal a direct interaction between NusA and the N-terminal Domain I of IF2, both highly abundant factors, providing a physical bridge between RNAP and the ribosome that can assist in transcription-directed translation initiation globally.

Available Cryo-EM structures of coupled RNAP and ribosomes which include NusA[78] show that the two machineries are oriented in a way compatible with IF2 binding to both the ribosome and NusA, if one considers that the IF2 Domain I is connected to the C-terminal part of IF2 via a long flexible linker[48] and that the coupled complex has a dynamic interface in the absence of NusG[83] (Supplementary Fig. 18). During the preparation of this manuscript, new studies have further uncovered new modes of TTC across different stages of gene expression[79,80,84]. Notably, a new class of complexes, termed transcription-translation complexes with long-range coupling (TTC-LC), has been identified[84]. These complexes exhibit a distinct architecture, characterized by their ability to accommodate extended mRNA linkers between RNA polymerase and the ribosome. TTC-LC structures have been proposed to function specifically during translation initiation, representing a mechanistically and spatially distinct mode of coupling. These studies highlight that coupling mechanisms are spatially and functionally diverse.

One of the most prominent roles of NusA in RNAP regulation is to stimulate hairpin-stabilized transcription pauses and termination[71,85]. Interestingly, RNAP pausing sites are enriched within the first 100 nt of expressed genes in both *E. coli* and *B. subtilis*[86]. We speculate that this transcriptome feature may be involved in the establishment of transcription-translation coupling, possibly through the interaction between IF2 and NusA, and play a regulatory role in gene expression. We propose a model where the frequent short-lived interactions

observed between NusA and IF2 might locally enrich IF2 near RNAP and thereby facilitate translation initiation. In addition, we detected long-lived, nucleoid-associated binding events specific to the IF2α isoform, suggesting that the NusA-IF2 interaction may also play a more direct role in coupling transcription and translation. Interestingly, the *nusA* and *infB* genes are part of an operon that is highly conserved throughout prokaryotic genomes[87], suggesting that the interplay between translation initiation and transcription regulation might be widespread among prokaryotes. However, neither the N-terminus of IF2 nor the C-terminus of NusA are conserved, and thus, further studies are needed to clarify if their physical interaction is common among different species.

Recent studies also suggest that NusA can promote liquid–liquid phase separation in bacterial cells and may contribute to the spatial organization of transcriptionally active regions within the nucleoid[59]. Furthermore, during the preparation of this manuscript, a new study reported that the N-terminal intrinsically disordered region of IF2 may also participate in the formation of biomolecular condensates, although this effect was prominent only under cold stress conditions[88]. While the relationship between such condensates and transcription or translation remains to be fully understood, these findings raise the intriguing possibility that condensates involving NusA and IF2 could act as hubs not only for transcription regulation but also for coordinating transcription and translation. Our data, however, do not address this question, as our microscopy experiments were performed on large ensembles of cells, making it difficult to detect condensates which lack determined spatial localization. Future studies using photo-activatable fluorophores to acquire enough data from individual cells may help to clarify the role of condensates in these processes.

Although the interaction between IF2 and NusA is dispensable for bacterial survival, it contributes to optimal growth in static conditions (Fig. 5a). More striking, however, is the dramatic effect on adaptation to poor growth conditions when the interaction is lost (Fig. 5b, c). For enteric bacteria like *E. coli*, the type and availability of nutrients can fluctuate significantly over time[89]. During metabolic transitions, such as a shift from the preferred carbon source glucose to a non-preferred source like succinate, cells experience rapid depletion of key metabolites[90]. Such nutrient stress triggers accumulation of the alarmone (p)ppGpp, which in turn drives global transcriptome reprogramming as part of the stringent response[91]. Following a carbon source shift, cells enter a lag phase before resuming growth, during which (p)ppGpp was shown to help coordinating transcription and translation[92]. Our data show that the interaction between NusA and IF2 is beneficial during recovery from the lag phase, potentially by contributing to the coordination of transcription and translation.

## Methods

### Bacterial strains and growth conditions
*Escherichia coli* K-12, strain MG1655, is referred to as the wild-type (WT) strain. For all molecular cloning procedures, Q5 High-Fidelity DNA Polymerase (NEB) was used for PCR amplification following the manufacturer's protocols.

**Plasmid construction.** To create plasmids for the inducible expression of initiation factors (IFs) and their fusion to HaloTag, a modified pQE-30 plasmid (QIAGEN), containing the *lacI* gene, was utilized. Genomic DNA from *E. coli* MG1655 served as template for amplification of the *infA*, *infB*, *infC*, *nusA*, and *rbfA* genes, as well as their respective fragments. Two approaches for plasmid construction were used: (i) Gibson Assembly for fragment insertion and (ii) PCR amplification of a plasmid with oligonucleotides modifying an original plasmid followed by re-circularization through PNK phosphorylation of the PCR product ends and ligation by T4 ligase (NEB). NEBuilder HiFi DNA Assembly Master Mix (NEB) was employed for Gibson Assembly, following the

manufacturer's protocols. The oligonucleotides used in this study are listed in Supplementary Data 57. Descriptions of cloning procedures for each plasmid constructed in this study are provided in Supplementary Data 58.

**Construction of strains containing chromosomal mutations.** For the construction of chromosomal mutations (deletions, and insertion of the gene for HaloTag), we employed λ Red assisted recombineering[93] via electroporation of linear DNA fragments obtained by PCR amplification and selection on LA plates containing required antibiotics and inducers. Oligonucleotides used to create the PCR products for recombineering are listed in Supplementary Data 57. Descriptions of strain constructions are provided in Supplementary Data 59. Strains *E. coli ΔompT*, *E. coli NusA-ΔC*, and *E. coli ΔIF2-Domain-I* had the resistance cassette removed using a helper plasmid pCP20 encoding the FLP recombinase[93].

**Doubling time measurements**

To measure the doubling time of strains lacking chromosomal genes for IFs with corresponding IFs or HaloTag-IFs fusion proteins present in pQE plasmids, bacterial strains were grown overnight on LB agar plates containing 40 µM isopropyl β-D-1-thiogalactopyranoside (IPTG) and 100 µg/ml ampicillin at 37 °C. For each strain tested, four individual colonies were grown in LB medium with 40 µM IPTG and 100 µg/mL ampicillin at 37 °C until they reached $OD_{600}$ of 0.1–1. The cells were then diluted to $OD_{600}$ of 0.0001 in fresh LB medium with varying concentrations of IPTG and growth kinetics were recorded using a BioTeK Synergy H1 plate reader in a 96-well microplate (200 µl per well) at 37 °C, with measurements taken at 5 minute intervals and shaking between measurements.

For measurements of doubling time of WT and *ΔIF2-Domain-I* strains in different media, the strains were grown on LB agar plates overnight at 37 °C. For each tested strain and medium, four individual colonies were grown for 24 hours in LB, M9 minimal medium supplemented with 0.4% glucose, or M9 minimal medium supplemented with 0.4% sodium succinate. The overnight cultures were diluted to $OD_{600}$ of 0.001 in the same fresh medium and the growth kinetics were recorded as described above. The doubling time during the exponential phase was calculated from OD measurements recorded by the plate reader in a 96-well microplate using a custom MATLAB script[94]. Data from four independent colonies were fitted separately and used to calculate the average doubling time and standard deviation for each strain and condition.

**Sample preparation for in vivo labeling using HaloTag**

*E. coli* cells containing HaloTag-fused molecules were grown overnight with shaking at 37 °C in 5 ml of LB medium containing necessary antibiotics. The cell culture was diluted 1:100 in fresh LB medium and grown to $OD_{600}$ of 0.5-1 at 37 °C with shaking. The cells were harvested by centrifugation at 4000 g for 5 min and washed twice with M9 medium with 0.4% glucose. The cells were resuspended in 150 µl of EZ Rich Defined Medium (RDM, Teknova) and JFX549 HaloTag ligand (a gift from Lavis lab) was added to the final concentration of 3 µM (0.2 µM for labelling of h6-MS2CP-HaloTag modified 30S). Cells were incubated at 25 °C for 30 min, washed twice with M9 medium with 0.2% glucose, and then incubated in RDM media at 37 °C with shaking for 40 min to facilitate the release of unbound ligand from the cells. After incubation, the cells were washed 3 times with M9 medium with 0.2% glucose, resuspended in RDM to OD600 of ≈0.003 and several 1 µl drops of the diluted cell suspension was placed on an agarose pad prepared with RDM and 2% agarose (SeaPlaque GTG Agarose, Lonza). The agarose pad was surrounded with a gene frame (Thermo Fisher) and sealed between the microscope slide and the coverslip (#1.5H, Thorlabs). The sample was placed on the microscope enclosed in an incubation chamber maintaining the temperature at 37 ± 2 °C. Cells

were grown for ≈70–120 min forming mini-colonies after which image acquisition was performed.

**Cell electroporation with Cy5-labeled molecules**

WT *E. coli* was used for electroporation of fMet-[Cy5]tRNA^fMet. Cells were grown overnight with shaking at 37 °C in 5 ml of LB medium. The cell culture was diluted 1:100 in 15 ml SOB medium without magnesium and grown to $OD_{600}$ of 0.6 at 37 °C with shaking. The cell culture was placed in ice bath for 20 min, then cells were made electrocompetent via 5 washing steps using 10% glycerol in ultrapure water (Milli-Q, Merck), and resuspended in 40 µl of 10% glycerol in ultrapure water. Competent cells (20 µl) were immediately mixed with 0.5 µl fMet-[Cy5] tRNA^fMet (4 µM) and electroporated in a 1-mm electroporation cuvette (Thermo Scientific) with a MicroPulser Electroporator (Bio-Rad) at a voltage of 1.9 kV. Cells were recovered from the cuvette by adding 1 ml of RDM and incubated for 30 min at 37 °C. Cells were washed 4 times with 0.5 ml RDM to remove non-internalized Cy5-labeled molecules via centrifugation at 2000 g followed by medium removal and resuspension in fresh medium. Cells were resuspended in RDM to $OD_{600}$ of ≈0.03 and several 1 µl drops of diluted cell suspension were placed on an agarose pad prepared with RDM and 2% agarose containing 1 µM SYTOX Blue (Invitrogen) for staining of dead cells. The agarose pad was surrounded with a gene frame and sealed between the microscope slide and the coverslip. The sample was placed on the microscope enclosed in an incubation chamber maintaining the temperature at 37 ± 2 °C. Cells were grown for ≈70–100 min forming mini-colonies after which image acquisition was performed.

**Optical Setup**

Widefield epifluorescence microscopy was performed on an inverted microscope (Nikon Ti2-E) equipped with a CFI Plan Apo lambda 1.45/100x objective (Nikon). The microscope was enclosed in an incubation chamber (H201-ENCLOSURE, OKOlab) with a temperature controller (H201-T-UNIT-BL, OKOlab) which maintained the temperature at 37 ± 2 °C. Phase contrast and bright-field images as well as fluorescence time-lapse movies were acquired with an Orca Quest camera (Hamamatsu). To track JFX549-coupled HaloTag molecules, a 546 nm laser (2RU-VFL-P-2000-546-B1R 2000 mW, MPB Communication) was used with the power density of 3 kW/cm² on the sample plane and stroboscopic mode of illumination with 3 ms laser pulses per 5 ms camera exposure. Imaging of Cy5-labelled molecules was performed with a 642 nm laser (VFL-P-1000-642-OEM5 1000 mW, MPB Communication) with a power density of 5 kW/cm² on the sample plane in stroboscopic illumination mode with 1.5 ms laser pulses per 5 ms camera exposure. Samples stained with SYTOX Blue were additionally imaged with a 405 nm laser (06-MLD 365 mW, Cobolt) with a power density of 10 W/cm² and continuous illumination with 21 ms exposure time. Between 200–1000 fluorescence images were acquired per minicolony to create a fluorescence time-lapse movie. The microscope was controlled using a µManager software package and acquisition of images from multiple positions was performed using custom µManager plugins.

**Microscopy data analysis**

Data analysis was performed using a custom MATLAB-based analysis pipeline. The cell segmentation was performed using phase contrast images using an adaptive local thresholding algorithm[95]. Segmentation masks were manually curated to remove poorly segmented cells and SYTOX-stained cells that died during electroporation (bright when illuminated with the 405-nm laser). Single fluorophores were detected in fluorescence images using the radial symmetry-based algorithm[96]. Refinement of positions of the fluorophores and estimation of position uncertainties was performed using symmetric Gaussian spot modelling and maximum posteriori fitting[37]. Two-dimensional trajectories of individual molecules were built by the uTrack algorithm. Trajectories

were built in segmented cells starting from the time point when there were ≤1 detected fluorophore per segmented cell in the current and the following frames and allowing gaps between detected positions (3 time points).

For each labeled molecule and experimental condition, at least three independent experiments were performed where each experiment included imaging of 30−100 positions with cell mini-colonies grown from single cells. Trajectories extracted from these experiments were analyzed using a previously described HMM-based algorithm[34,37]. Briefly, all detected trajectories longer than 5 frames were fitted to a pre-defined number of discrete diffusion states, characterized by diffusion coefficient, occupancy, and dwell time. The resulted models were coarse-grained to 2-3-state models based on clustering of diffusion states using a threshold at $1\,\mu m^2/s$, or two thresholds at $1\,\mu m^2/s$ and $8\,\mu m^2/s$, to separate "Slow diffusion state" and "Fast diffusion state", and cleavage products. Occupancies and dwell-times presented in bar plots are calculated from coarse-grained 5-state HMM models, with averages calculated from independent experiments. Error bars represent standard deviations between these independent experiments. For all combined datasets with HaloTag-labeled molecules where kinetic data were used to make conclusions, we collected at least 100,000 trajectory steps, for other datasets we collected at least 50,000 trajectory steps.

For the spatial occupancy heatmap plots we performed cell sorting to account for variability in nucleoid number and positioning, which can differ across cell growth stages and even among cells at similar stages of division[52]. We excluded newly divided cells−characterized by a single nucleoid at mid-cell or two poorly separated lobes −and larger cells that often contain more than two nucleoid lobes[52]. Data without coarse-graining were used to plot spatial distributions of 30 S subunits and RNAP. Coarse-grained 5-state HMM models with a threshold at $1\,\mu m^2/s$ were used for plotting of IF2-Domain-I-HaloTag, NusA-HaloTag, HaloTag-IF2α, HaloTag-IF2γ. For RbfA-HaloTag a threshold of $1.2\,\mu m^2/s$ was applied, as its main slow diffusional states exhibit diffusion coefficients in the range of $1–1.2\,\mu m^2/s$. For HaloTag-IF2α-S753Y and HaloTag-IF2γ-S753Y, additional heatmaps using thresholds of $0.25\,\mu m^2/s$ and $0.05\,\mu m^2/s$ were generated to illustrate tendencies in the spatial distribution of the slow diffusion states. Statistical significance between groups was assessed using a two-sided unpaired *t*-test. *P*-values are indicated in the figure as follows: *P*<0.05 (*), *P*<0.01 (**), *P*<0.001 (***), and not significant (ns) otherwise.

### Pull-down assay

*E. coli ΔompT nusA-HaloTag-kan* strain and *E. coli ΔompT* strain carrying pColA-HaloTag plasmid were transformed with plasmids pQE-6His-IF2α, pQE-6His-IF2γ, or pQE-IF2-Domain-I-6His and plated on LB agar plates containing 50 μg/mL kanamycin and 100 μg/mL ampicillin and grown overnight at 37 °C. Several colonies of each strain were used to inoculate 10 ml LB media containing 50 μg/mL kanamycin and 100 μg/mL ampicillin and grown overnight at 37 °C. The overnight cultures were diluted 1:100 in 400 mL of fresh LB with antibiotics and grown to $OD_{600}$ of ≈0.6 at 37 °C after which IPTG was added to 0.5 mM final concentration to induce production of IF2 variants. Cells were cooled down in ice bath and harvested by centrifugation (4,000 g, 15 min, 4 °C) four hours after induction by IPTG. Cell lysis and affinity purification were performed at 4 °C or on ice.

The cell pellets were resuspended in 10 ml of Lysis buffer (20 mM Tris-HCl pH 7.5, 200 mM NaCl, 5 mM $MgCl_2$, 5% glycerol) supplemented with phenylmethylsulfonyl fluoride (PMSF, Sigma) and cOmplet Protease Inhibitor Cocktail (Roche) and 1 mg/ml lysozyme (Merck), incubated for 1 h on ice and disrupted by sonication (Sonics® Vibra-Cell VCX 130) (3 min cycle per sample: pulse 10 s, 20 s off, 50% amplitude). Cell debris was removed by centrifugation (10,000 g, 30 min, 4 °C). An aliquot of the lysate (20 μl) was mixed with JFX549 dye (1 μM final concentration), incubated at room temperature for

20 min and then mixed with an equal volume of 2x Laemmli Sample Buffer (Bio-Rad) containing BME and denatured at 98 °C for 5 min. The rest of the lysates were loaded in a gravity column containing 500 μl of HisPur cobalt resin (ThermoFisher Scientific). The columns were washed with 75 ml of wash buffer 1 (20 mM Tris-HCl pH 7.5, 200 mM NaCl, 5 mM $MgCl_2$, 5% glycerol, imidazole 20 mM) and the bound proteins were eluted using 1 ml of elution buffer 1 (20 mM Tris-HCl pH 7.5, 200 mM NaCl, 5 mM $MgCl_2$, 5% glycerol, imidazole 300 mM). An aliquot of the elution fraction (20 μl) was mixed with JFX549 dye (1 μM final concentration), incubated at room temperature for 20 min after which an equal volume of 2x Laemmli Sample Buffer containing BME was added and samples were denatured at 98 °C for 5 min. For SDS-PAGE 2 μl of lysates, 10 μl of elution fractions, and 1 μl of PageRuler Plus Prestained Protein Ladder (ThermoFisher Scientific) were loaded on a Mini-Protean TGX gel (4−20 %, Bio-Rad) and gel electrophoresis was performed according to manufacturer's protocol. The JFX549 bound proteins were visualized using ChemiDoc MP imaging system (Bio-Rad).

### BPA cross-linking

*E. coli ΔompT nusA-HaloTag-kan* strain was transformed with pEVOL-pBpF (Addgene, #31190) and the plasmid pQE-IF2-Domain-I-6His or its derivatives for BPA incorporation (pQE-IF2-DomainI-BPA13-6xHis, pQE-IF2-DomainI-BPA50-6xHis, and pQE-IF2-DomainI-BPA88-6xHis) and plated on LB agar plates containing 50 μg/ml kanamycin, 34 μg/ml chloramphenicol, and 100 μg/mL ampicillin and grown overnight at 37 °C. Several colonies of each strain were used to inoculate 10 ml LB media containing 34 μg/ml chloramphenicol, and 100 μg/mL ampicillin and grown overnight at 37 °C. The overnight cultures were diluted 1:100 in 25 mL of fresh LB with corresponding antibiotics, grown to $OD_{600}$ of 0.6, and then mixed with 25 ml of LB media containing 10 mM arabinose, 1 mM IPTG, 2 mM p-benzoyl-L-phenylalanine (BLD pharm, Bpa), 34 μg/ml chloramphenicol, and 100 μg/mL ampicillin. Induced cells were grown for 4 h at 37 °C after which 25 ml of the cell culture was exposed to $UV_{365}$ for 12 min at room temperature (illumination using UVP 2UV transilluminator, Analytik Jena) and the rest of the culture (25 ml) was kept at room temperature without UV irradiation. Cells were cooled down in ice bath and harvested by centrifugation (4000 g, 15 min, 4 °C).

The cell pellets were resuspended in 2 ml of B-PER bacterial protein extraction reagent (ThermoFisher Scientific) supplemented with cOmplet Protease Inhibitor Cocktail and lysed at room temperature for 15 min. Cell debris was removed by centrifugation (17,000 g, 5 min). The cell lysates were mixed with 50 μl of HisPur cobalt resin and incubated with shaking for 1 h. The resin was washed 4 times with wash buffer 2 (20 mM Tris-HCl pH 7.5, 250 mM NaCl, imidazole 20 mM) via centrifugation (1,000 g, 2 min) and resuspension in fresh wash buffer 1. Proteins bound to the HisPur cobalt resin were eluted using 100 μl of elution buffer 2 (20 mM Tris-HCl pH 7.5, 250 mM NaCl, imidazole 300 mM). An aliquot of the elution fraction (20 μl) was mixed with JFX549 dye (1 μM final concentration), incubated at room temperature for 20 min after which an equal volume of 2x Laemmli Sample Buffer containing BME was added and samples were denatured at 98 °C for 5 min. For SDS-PAGE, 10 μl of elution fractions and 1 μl of PageRuler Plus Prestained Protein Ladder were loaded on a Mini-Protean TGX gel (4-20%) and gel electrophoresis was performed according to the manufacturer's protocol. The JFX549 bound proteins were visualized using ChemiDoc MP imaging system.

### Growth experiments for WT and ΔIF2-Domain-I *E. coli* strains

WT and *ΔIF2-Domain-I* strains were grown overnight on LB agar plates at 37 °C. Four individual colonies of each strain were used to inoculate 3 ml of LB media, M9 medium supplemented with 0.4% glucose, and M9 medium supplemented with 0.4% sodium succinate. The cultures were grown for 24 h at 37 °C with shaking. From each culture, 1 ml was

harvested by centrifugation at 4000 g for 5 min, and cells were washed twice with M9 medium without a carbon source. The cells were then resuspended in fresh medium to $OD_{600}$ of 0.05. Growth kinetics were recorded using a BioTeK Synergy H1 plate reader in a 96-well microplate (200 μl per well) at 37 °C, with measurements taken at 5-minute intervals and shaking between measurements.

## Reporting summary

Further information on research design is available in the Nature Portfolio Reporting Summary linked to this article.

## Data availability

The microscopy data generated in this study have been deposited in the SciLifeLab Repository: https://doi.org/10.17044/scilifelab.28795805. Source data are provided with this paper. All unique biological materials are available from the corresponding author upon request.

## Code availability

The computational code used for analysis and plotting is available from the corresponding author and in the SciLifeLab Repository: https://doi.org/10.17044/scilifelab.28795805.

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

## Acknowledgements

The authors would like to thank the laboratory of Luke Lavis for providing the JFX549-HaloTag ligand, Suparna Sanyal and Xueliang Ge for discussions, and Irmeli Barkefors for comments on the manuscript. MATLAB scripts for plotting growth curves were developed with the assistance of ChatGPT. This work was supported by the European Research Council (947747-SMACK), and the Swedish Research Council (2019-03714, 2023-03383).

## Author contributions

M.J. conceived the project. M.M. developed the labeling scheme. M.M. and M.J. designed the experiments. M.M. performed all cloning and all experiments, and wrote code for visualization of the data. M.M. and M.J. wrote the paper.

## Funding

## Competing interests

The authors declare no competing interests.
