## [Transparent Peer Review file · Nature Communications]

A complex between IF2 and NusA suggests early coupling of transcription-translation

Corresponding Author: Dr Magnus Johansson

Version 0:

Reviewer comments:

Reviewer #1

(Remarks to the Author)

In this work, Metelev and Johansson interrogate the *in vivo* dynamics with which the bacterial translation initiation factors IF1, IF2, and IF3 interact with the ribosomal small, 30S, subunit, impressively allowing them to deduce the *in vivo* cascade through which a 30S pre-initiation complex (PIC) assembles. Notably, the *in vivo* dynamics they observe are consistent with decades of *in vitro* biochemical and biophysical studies of this process and, among other findings, reveal that the opposing roles IF2 and IF3 have been observed to play in *in vitro* studies of subunit joining are also observed *in vivo*. The authors also characterize the dynamics of IF2 α , a physiologically important isoform of IF2 that has been relatively neglected in *in vitro* biochemical studies of bacterial translation initiation. By expanding on previously published pull-down studies, the authors combined site-directed mutagenesis with both single-particle tracking microscopy and additional pull-downs to demonstrate that the N-terminal domain that is unique to the IF2 α isoform binds to NusA, a protein with a role in physically connecting RNA polymerase to ribosomes as a part of coupling transcription to translation in bacteria. Based on this finding, the authors propose that a direct IF2 α -NusA bridge may couple transcription to 30S PIC assembly.

The studies reported by Metelev and Johansson are technically sound, solid genetic and biophysical work. Their results agree with the findings of decades of *in vitro* genetic and biochemistry work. Furthermore, the direct interaction they demonstrate between the N-terminal domain of IF2 α and NusA is thought-provoking. Despite the thought-provoking nature of this observation, it is only the starting point for determining whether and how an IF2-NusA complex couples transcription to translation initiation. The authors do not substantially investigate the contribution that the interaction between IF2 α and NusA makes to transcription-translation coupling, and the data presented in this manuscript do not currently support their conclusion that an IF2-NusA complex links transcription and translation. Specifically, the authors do not demonstrate that such a complex forms or plays a role during translation initiation. Consequently, the findings reported in this manuscript, as they currently stand, would be better suited for a journal with a more specific focus on translation or transcription than one with the broad readership of Nature Communications.

Reviewer #2

(Remarks to the Author)

Metelev and Johansson present a comprehensive study on the functional roles of translation initiation factors IF3 and IF2, including its isoforms IF2 α and IF2 β , as well as initiator tRNA (fMet-tRNA^{fMet}) within live *E. coli* cells. Using powerful single-particle tracking approaches, they observe the diffusion of a range of HaloTag-labelled proteins involved in transcription and translation, corroborating these observations through genetic perturbations and biochemical assays. A significant finding is the detection of a direct interaction between the N-terminal Domain I of IF2 α and the C-terminal domain of NusA, suggesting a hitherto unknown mechanism for transcription-translation coupling during translation initiation. The *in vivo* crosslinking experiments provide strong evidence for the interaction, and the use of alpha-fold 3 in selecting the sites makes the approach very timely. Overall, this study offers novel insights into the functional role of IFs, while reinforcing previously established findings through advanced single-molecule techniques. For instance, the authors identified that initiator tRNA is last to arrive at the ribosome based on its dwell time, consolidating previous *in vitro* discoveries.

The article is well-structured and provides a good overview of the biological background and maintains clear motivation throughout. Some additional background and schematics will help make the article understandable to a broader audience. The methodology is detailed without being superfluous, and therefore experiments will be reproducible. Overall, the study offers valuable insights into bacterial initiation factor dynamics; however, some limitations were identified that should be

considered.

Major points:

1. The authors suggest that the complex between IF2 and NusA links transcription and translation, – however, the conclusion that IF2 α couples translation to transcription is more tentative and not sufficiently pursued and supported. Additional experiments are needed to substantiate functional coupling, as the in vitro pull-down and cross-linking assays do not verify coupled activity, and current data based on bacterial growth analysis shows broad cellular disruption. Techniques that show specific disruption of transcription rate and/or translation initiation would strengthen this conclusion. Alternatively, the main text, abstract and title need to be revised to indicate that there is no direct evidence of IF2 α acting as a bridge, and that (while this is a reasonable suggestion) this needs to be further tested by future work.
2. Can the authors exclude that NusA binds to a ribosome that is translating a nascent mRNA, but is not coupled to the RNA polymerase?
3. Although replicate experiments were conducted, the lack of experimental repeats limits the robustness of the conclusions. On a similar vein, there is a lack of statistical tests to assess the significance of differences in state occupancy and dwell time between the WT and mutant strains.
4. The rationale for using a large number of states for fitting the tracking results is unclear. Why not fit fewer states (especially given the absence of enough S/R to warrant such a large number of states)? In most cases, it appears that a 2- or 3-state model will be sufficient for the comparisons. It is also unclear how well the data are fit by the fitted parameters. Finally, how robust are the main conclusions with respect to the number of states used?
5. A definition of “immature” and “mature ribosomes” should be provided to help understand claims and interpretations and to clarify why the authors assume that immature ribosomes are tethered, and specifically via RNAP (and not via nascent RNA).
6. Often in the manuscript there is a mention for 30S binding, but it is unclear whether binding to the 30S occurs while the 30S is on or off the mRNA. This is important to establish, since if the binding occurs on a 30S ribosomal subunit that is not engaged with the RNA, there is no connection to the translation cycle since the step that may be affected precedes association with the nascent RNA. The authors should also comment on the possibility that IF2 α enters the RNA as a complex with another protein or even the 30S subunit.
7. NusA is also a major antitermination factor, with high occupancy on rRNA operons. The results interpretation and discussion need to account for this. What prevents IF2 α from binding there or from binding to NusA in the cytoplasm? Further, the ms needs to discuss/cite in the putative formation of NusA condensates (see Ref. 57), the involvement of NusA as the putative bridging factor in TTC in *M. pneumoniae* cells (<https://pmc.ncbi.nlm.nih.gov/articles/PMC7115962/>), and the Mooney et al ChIP-chip work on NusA (<https://doi.org/10.1016/j.molcel.2008.12.021>).
8. Pg 14: The results show that NusA interacts with the chromosome (which is expected) and that IF2 α interacts with the chromosome as well. This result is one of correlation/colocalization, and does not prove an interaction; this distinction should be clarified in the ms.
9. A more detailed explanation for the rationale behind projecting only the middle 40% of dot coordinates is required – to what extent is the data affected by this choice? This is also important in the context of the RNAP distribution in vivo, where it has been shown that transcribing polymerases are in the periphery of the nucleoid (<https://doi.org/10.1073/pnas.1507592112>).
10. The authors allude to the NusA-IF2 mediated coupling contributing to a rewiring of the transcriptome and translome without any analysis of gene expression.
11. To improve clarity, the authors could extend the figures in the main text, e.g. incorporate Supplementary Figure 6 into Figure 3, to reduce redundancy and prevent flicking back-and-forth. Additionally, they could include the sample sizes (n values) into the main text, rather than in the supplementary material. A schematic of translation initiation would have been helpful in the intro (include in Fig 1), further, a schematic diagram of their proposed model may improve audience understanding.
12. The article may benefit from expanding on the relevance of adaptation mechanisms to changing growth conditions to further the discussion and highlight the importance of their findings.

Minor

1. The figures need improvement; the font sizes and some of the figure elements are too small.
2. An example of the HMM fits for at least one factor needs to be in the main text to showcase the approach and its limitations.
3. The length of the trajectories needs to be discussed since it does provide an indication of the observation span and the timescales of transitions that can be observed.

Reviewer #3

(Remarks to the Author)

Reviewer #4

(Remarks to the Author)

The manuscript "A complex between IF2 and NusA links transcription and translation" by Metelev and Johansson presents a plethora of experiments leveraging cutting-edge microscopy technique to probe into the behaviour and interaction of the translation initiation factors in live *E. coli* cells. Particularly interesting is the newly found connection between the lack and altered functionality of specific IFs and the adaptation of *E. coli* to new growth conditions in the form of prolonged lag phases. This study convincingly shows how single-particle tracking (spt) can enrich the study of biological systems in situ. Overall, this is a very well written and technically strong manuscript in which the authors included many controls to back up their biological statements. In the following, I want to focus a bit on the technical aspects of the study, which I hope can be addressed by the authors.

Major comments

- As mentioned, the authors performed many spt experiments in many different genetic backgrounds. If I understand correctly, rather than using conventionally calculated (apparent) diffusion coefficients to infer the mobility of complexes, the authors fed the localisations and tracks into a hidden Markov model (HMM) to obtain dwell times and diffusion coefficients. Whereas overall some of the obtained results are in line with previous reports, the current version of the manuscript makes it difficult to judge for the reader how well the occupancies obtained by HMM describe the raw data obtained through single-particle tracking. To this end, I would suggest generating more conventional diffusional distributions of their tracks and provide them as supplementary material. As such, a qualitative evaluation of the diffusional equilibrium can be used to gauge whether the relative occupancies in slow and fast states provided by the HMM fitting are actually a good interpretation. Along the same lines, and maybe the authors have done that already, a comparison of the HMM output from the individual repeats (all measurements were at least measured in triplicates) could help to build further trust into the HMM data. As of now, the HMM, albeit well-described in literature, is a black box that will always provide output with the challenge being the correct interpretation.

- Regarding the statistics, maybe rather than just talking of microcolonies that were imaged, could the authors state the number of cells imaged per condition and the number of tracks per condition?

- The use of HaloTag enables the generation of long tracks, reaching even second-long durations. The authors combined the HaloTag with fast exposures and frame times, being able to also resolve interactions as short as 20 ms within their tracks. At the same time, the HMM applied here to fit their single-particle tracking data seems to account for a balance between slow and fast states of the imaged proteins at the steady-state. Does this approach exclude the possibility that the same protein changes of state while it is being visualised (within the same track), which could be the case for some conditions in this manuscript? If state transitions are indeed a common occurrence due to track length and average dwell times, how is their HMM approach resistant to this issue? I would further suggest including some example histograms of track lengths to get a better idea how long the average track can be.

Minor comments.

- In the introduction, the authors mention the different isoforms of IF2. It is however unclear to me whether they come from different genes or whether other processes are involved in the generation of IF2₁, IF2₂ and IF2₃ from the same *infA* gene. Could the author add a brief explanation in the introduction section on how the isoforms are generated? In the abstract, the authors refer to two naturally occurring isoforms, so that is a bit confusing or am I missing something?

- Page 4. Fig.1 / Second paragraph results section and throughout. Please indicate whether IF was tagged on the chromosome or was made available on a plasmid.

- Page 11. In the sentence "Quite remarkably, a single mutation, S753Y, renders IF2 to preferentially adopt the active conformation independent of fMet-tRNA^{fMet} 24,45." there seem to be a verb missing, the sentence does not make sense as is.

- Page 14. In the sentence "As an initial naïve test to find the additional binding partner of IF2₂, we decided to...", IF2₂ is supposed to be IF2_α.

- Page 14. In the "The N1 domain of IF2 binds to NusA" Results section, the fusion of RbfA and HaloTag is first spelled as RbfA-HaloTag and then as HaloTag-RbfA. Often the spelling of the fusions indicates at what terminus the proteins are fused. Could the author provide a consistent naming, also consistent with the terminus at which the HaloTag is fused to?

- Page 15. "As mentioned previously,... previously" Delete one instance. Add reference here.

- Page 25. "model sizes 6-9" What does that mean?

- In the sample preparation sections of Materials and Methods, cells are diluted to a final OD before being deposited on agarose pads to form microcolonies. In the section "Sample preparation for in vivo labeling using HaloTag" the final OD value is 0.003, whereas in the section "Cell electroporation with Cy5-labeled molecules" the final OD value is 0.03. Are the two final OD values actually

Version 1:

Reviewer comments:

Reviewer #2

(Remarks to the Author)

The authors have taken on board the comments and made many changes in the text to clarify their findings; they also added new results and analysis. In particular, the authors have (for the most part) appropriately adjusted their conclusions to suggest a potential role of IF2a-NusA in coupling transcription and translation rather than a direct linkage. However, there are still some instances where the language has not been corrected, e.g. the title remains unchanged, and see line 78: "Hence, our results suggest a hitherto unknown mechanism for transcription-translation coupling during translation initiation." Toning down these statements before publication will be appropriate.

Reviewer #3

(Remarks to the Author)

Reviewer #4

(Remarks to the Author)

The authors addressed all my concerns and suggestions. By adding more explanation on the HMM data analysis, comparisons with conventional step sizes distributions, and more statistical rigour the manuscript now makes an (other) excellent case for using single-molecule resolved imaging to elucidate biological functions and pathways.

NCOMMS-24-65154-T

Response to reviewers' comments

Dear Dr Johansson,

Thank you again for submitting your manuscript "A complex between IF2 and NusA links transcription and translation" to Nature Communications. We have now received reports from 4 reviewers and, based on their comments, we have decided to invite a revision of your work. Your revision should address all the points raised by our reviewers (see their reports below).

When resubmitting, you must provide a point-by-point response to the reviewers' comments. Please show all changes in the manuscript text file with track changes or colour highlighting. If you are unable to address specific reviewer requests or find any points invalid, please explain why in the point-by-point response.

Response:

We thank the editor and the reviewers for their constructive feedback, which we think have improved the quality of the manuscript considerably. We hope and believe that we have been able to address all the concerns. Most importantly, we have performed additional experiments supporting our conclusions, and we have revised our data analysis to provide a more transparent and unbiased representation of the results and their reproducibility.

REVIEWER COMMENTS

Reviewer #1 (Remarks to the Author):

In this work, Metelev and Johansson interrogate the in vivo dynamics with which the bacterial translation initiation factors IF1, IF2, and IF3 interact with the ribosomal small, 30S, subunit, impressively allowing them to deduce the in vivo cascade through which a 30S pre-initiation complex (PIC) assembles. Notably, the in vivo dynamics they observe are consistent with decades of in vitro biochemical and biophysical studies of this process and, among other findings, reveal that the opposing roles IF2 and IF3 have been observed to play in in vitro studies of subunit joining are also observed in vivo. The authors also characterize the dynamics of IF2 α , a physiologically important isoform of IF2 that has been relatively neglected in in vitro biochemical studies of bacterial translation initiation. By expanding on previously published pull-down studies, the authors combined site-directed mutagenesis with both single-particle tracking microscopy and additional pull-downs to demonstrate that the N-terminal domain that is unique to the IF2 α isoform binds to NusA, a protein with a role in physically connecting RNA polymerase to ribosomes as a part of coupling transcription to translation in bacteria. Based on this finding, the authors propose

that a direct IF2 α -NusA bridge may couple transcription to 30S PIC assembly.

The studies reported by Metelev and Johansson are technically sound, solid genetic and biophysical work. Their results agree with the findings of decades of in vitro genetic and biochemistry work. Furthermore, the direct interaction they demonstrate between the N-terminal domain of IF2 α and NusA is thought-provoking. Despite the thought-provoking nature of this observation, it is only the starting point for determining whether and how an IF2-NusA complex couples transcription to translation initiation. The authors do not substantially investigate the contribution that the interaction between IF2 α and NusA makes to transcription-translation coupling, and the data presented in this manuscript do not currently support their conclusion that an IF2-NusA complex links transcription and translation. Specifically, the authors do not demonstrate that such a complex forms or plays a role during translation initiation. Consequently, the findings reported in this manuscript, as they currently stand, would be better suited for a journal with a more specific focus on translation or transcription than one with the broad readership of Nature Communications.

Response 1.0:

We thank the reviewer for an overall positive evaluation of our work. We agree that this manuscript doesn't give all the answers regarding the functional role of the IF2 α -NusA interaction and the mechanism of coupling between RNAP and the ribosome. However, we are convinced that this work will become a launchpad for further research, including structural studies, deciphering the functional role and evolutionary role for the highly conserved *nusA-infB* gene pair. We believe that such further studies require a carefully designed reconstituted system which is beyond the scope of our manuscript. That said, we agree with the reviewer, that additional data which show that an RNAP-ribosome coupled complex is formed would strengthen the manuscript. Therefore, we tried to address this point and performed additional experiments to provide further evidence of formation of the coupled complex.

In the first version of the manuscript we provided strong evidence that the N-terminal Domain I, present only in the IF2 α isoform, interacts with NusA. This was supported by cross-linking and pull-down experiments. Furthermore, we showed that IF2 Domain I exhibits short, transient bindings to NusA, and that its spatial distribution suggests interaction with NusA already bound to RNA polymerase. However, we did not sufficiently investigate whether such transient interactions can lead to formation of a higher-order complex containing all four components, RNAP:NusA:IF2 α :ribosome. In principle, if such a complex is formed, it should be present in our microscopy data. However, two factors complicate its detection in WT cells with WT IF2 α factor: i) WT IF2 α often interacts with both the ribosome and NusA in a transient manner, making detection and interpretation challenging; ii) translation initiation events that are facilitated by RNAP-NusA interactions represent only a small subset of all initiation events, and only the leading ribosome is expected to form such a complex. To address the first limitation, we performed additional tracking experiments of the two IF2 isoforms, now containing a previously described S753Y mutation which stabilizes the factor's interaction with the ribosome. We also used a slower acquisition rate (30 ms per frame) to limit the detection of transient interactions between NusA and IF2. To address the second limitation and to enrich a potential fraction of coupled complexes, we performed these tracking experiments in a strain where the IF2 α isoform is not present, thus eliminating competition from endogenous IF2 α . Analysis of spatial distributions of long

lasting binding events in these additional experiments show a strong isoform-specific effect, further supporting a model in which Domain I of IF2 assists in establishment of coupling between the RNAP and the ribosome. We believe that these experiments can help to design an experimental setup for structural analysis of coupled complexes which would, in the future, validate our findings. Please see these new experiments in the new section of the manuscript “Isoform-specific dynamics of IF2 in the bacterial nucleoid”.

Furthermore, we acknowledge in the revised manuscript that with our microscopy techniques, it is inherently challenging to prove multifactor interactions, and that confirming the existence and nature of such complexes will require further structural studies. That said, we note that other approaches, such as structural methods, are not without limitations and may introduce artifacts due to the *in vitro* conditions. In contrast, our data provide *in vivo* support for the likelihood that such complexes can form under physiological conditions.

Reviewer #2 (Remarks to the Author):

*Metlev and Johansson present a comprehensive study on the functional roles of translation initiation factors IF3 and IF2, including its isoforms IF2a and IF2g, as well as initiator tRNA (fMet-tRNA^{fMet}) within live E. coli cells. Using powerful single-particle tracking approaches, they observe the diffusion of a range of HaloTag-labelled proteins involved in transcription and translation, corroborating these observations through genetic perturbations and biochemical assays. A significant finding is the detection of a direct interaction between the N-terminal Domain I of IF2a and the C-terminal domain of NusA, suggesting a hitherto unknown mechanism for transcription-translation coupling during translation initiation. The *in vivo* crosslinking experiments provide strong evidence for the interaction, and the use of alpha-fold 3 in selecting the sites makes the approach very timely. Overall, this study offers novel insights into the functional role of IFs, while reinforcing previously established findings through advanced single-molecule techniques. For instance, the authors identified that initiator tRNA is last to arrive at the ribosome based on its dwell time, consolidating previous *in vitro* discoveries.*

The article is well-structured and provides a good overview of the biological background and maintains clear motivation throughout. Some additional background and schematics will help make the article understandable to a broader audience. The methodology is detailed without being superfluous, and therefore experiments will be reproducible. Overall, the study offers valuable insights into bacterial initiation factor dynamics; however, some limitations were identified that should be considered.

Response 2.0:

We thank the reviewer for the thorough evaluation of our work and the positive assessment. We find the points raised to be highly valuable and have carefully addressed all of them in the revised manuscript. In particular, we have substantially revised the data presentation and our data analysis to address concerns regarding reproducibility. We have also expanded explanations throughout the manuscript to improve clarity and significantly extended the discussion section to incorporate broader context, including new studies published during the preparation of this

manuscript. Furthermore, we performed additional experiments that further strengthen the conclusions presented in the original version.

Major points:

1. The authors suggest that the complex between IF2 α and NusA links transcription and translation, – however, the conclusion that IF2 α couples translation to transcription is more tentative and not sufficiently pursued and supported. Additional experiments are needed to substantiate functional coupling, as the in vitro pull-down and cross-linking assays do not verify coupled activity, and current data based on bacterial growth analysis shows broad cellular disruption. Techniques that show specific disruption of transcription rate and/or translation initiation would strengthen this conclusion. Alternatively, the main text, abstract and title need to be revised to indicate that there is no direct evidence of IF2 α acting as a bridge, and that (while this is a reasonable suggestion) this needs to be further tested by future work.

Response 2.1:

We thank the reviewer for this important point. Also the other reviewers noted that our original data did not sufficiently support the claim that IF2 α forms a physical bridge between the ribosome and RNAP via its interaction with NusA. It is indeed possible that the interaction between NusA and IF2 α may instead serve to locally enrich IF2 near RNAP through transient contacts, or have some other regulatory role. In the revised manuscript, we acknowledge that with our microscopy techniques, it is inherently challenging to prove multifactor interactions, such as a higher-order complex involving all four components: RNAP, NusA, IF2 α , and the ribosome. We also explicitly acknowledge that establishing the existence of such a complex will require complementary structural studies in the future. Nevertheless, we tried to address this point and performed additional experiments to provide further evidence of formation of the higher-order complex.

In the original version of the manuscript, we showed that IF2 Domain I exhibits short, transient binding to NusA, and that its spatial distribution suggests interaction with NusA already bound to RNA polymerase. However, we agree that these experiments do not demonstrate the formation of a higher-order complex that also includes the ribosome. Proving the existence of such a complex in vivo is challenging. For instance, in vivo cross-linking involving multiple interactors cannot rule out sequential, rather than simultaneous, interactions.

We hypothesized that if the higher-order complexes form, they might be detectable by HMM analysis, as they would show even slower diffusion due to association with the chromosome. However, detecting them in WT cells with native IF2 α is difficult because: (i) IF2 α often interacts both with the ribosome and NusA in a transient manner, complicating interpretation; and (ii) translation initiation mediated by RNAP–NusA occurs in only a small subset of events, involving only the leading ribosome. To address these limitations, we performed new tracking experiments using both IF2 isoforms carrying the previously described S753Y mutation, which stabilizes their interaction with the ribosome. We also used a slower acquisition rate (30 ms per frame) to reduce the detection of transient NusA–

IF2 interactions. To enrich for potential coupled complexes, these experiments were performed in a strain lacking endogenous IF2 α , thereby eliminating competition and increasing the availability of unoccupied RNAP–NusA.

Analysis of spatial distributions of long lasting binding events in these additional experiments show a strong isoform-specific effect, further supporting a model in which Domain I of IF2 assists in establishment of coupling between the RNAP and the ribosome. Please see these new experiments in the new section of the manuscript “Isoform-specific dynamics of IF2 in the bacterial nucleoid”.

Finally, we have toned down our conclusions, as we fully agree with the reviewer that further studies are needed to validate the findings of our work.

2. Can the authors exclude that NusA binds to a ribosome that is translating a nascent mRNA, but is not coupled to the RNA polymerase?

Response 2.2:

We thank the reviewer for this great question which helped us design a new valuable experiment. Indeed, spatial distribution analysis of slowly diffusing NusA indicate that it predominantly associates with the chromosome, but we cannot exclude that some fraction interacts solely with the ribosome or with IF2 α in an RNAP-independent manner. Since RNAP binding of NusA is primarily mediated by its N-terminal domain (NTD), we have performed additional experiments to investigate how an NTD deletion affects NusA diffusion. Comparing HMM results for Δ NTD-NusA-HaloTag in WT cells and in cells lacking IF2 α show that NusA can interact with IF2 α , but no IF2-independent interaction with the ribosome was detected. In other words, these experiments show that NusA can associate with ribosomes independently of its interaction with RNAP. However, this ribosome-associated population is more than one order of magnitude less abundant than the RNAP-bound population and is strongly dependent on the presence of the IF2 α isoform. These new results are included in the revised manuscript in the last paragraph of section “The N1 domain of IF2 binds to NusA”.

3. Although replicate experiments were conducted, the lack of experimental repeats limits the robustness of the conclusions. On a similar vein, there is a lack of statistical tests to assess the significance of differences in state occupancy and dwell time between the WT and mutant strains.

Response 2.3:

We thank the reviewer for pointing out this issue. We have revised our analysis and data presentation to explicitly compare individually performed replicate experiments. In the original version of the manuscript, the combined datasets for different molecules varied significantly in total number of trajectory steps. Therefore, we have now performed additional replicate experiments to ensure that we have >100,000 trajectory steps for all datasets with HaloTag-labeled molecules where kinetic data were used to make conclusions. We also evaluated the statistical

significance between groups using a standard two-sided unpaired t-test and included those in the manuscript. Although the conclusions from the original manuscript remain valid, our new analysis shows that IF2 α responds insignificantly to the mutagenesis, in contrast to IF2 γ .

4. The rationale for using a large number of states for fitting the tracking results is unclear. Why not fit fewer states (especially given the absence of enough S/R to warrant such a large number of states)? In most cases, it appears that a 2- or 3-state model will be sufficient for the comparisons. It is also unclear how well the data are fit by the fitted parameters. Finally, how robust are the main conclusions with respect to the number of states used?

Response 2.4:

Indeed, in our analysis we would prefer to work with simple models, as they are easier to interpret. However, we often find that the microscopy data are complex. Long trajectories obtained using HaloTag allow us to track molecules over extended periods, and even visual inspection reveals underlying heterogeneity in diffusion behavior. In our previous work on ribosome tracking, we observed diffusion rates spanning at least two orders of magnitude, likely reflecting biologically meaningful states. For example, the slowest-diffusing ribosomes are typically located near the membrane and likely tethered via translocons, while freely diffusing subunits move much faster. This complexity for the ribosome diffusion is seen even in more simple analysis such as analysis of step-length distributions (see new Supplementary Fig. 5). In our HMM analysis, to remain unbiased, we do not enforce any particular model size or pre-set model parameters, such as diffusion coefficients for diffusion states, that we would like to look at. As a result, the number and nature of detected states depend on the complexity and abundance of biologically relevant behaviors, which tend to emerge as model size increases. Statistical tools such as the Akaike Information Criterion (AIC) consistently favor larger models when applied to real microscopy data (see Supplementary Fig. 6). On the contrary, in our experience, HMM analysis of simulated data, where diffusion parameters are well defined, often favors simpler models. Given this, our approach is to examine larger HMM models and then coarse-grain the results into biologically relevant states, based on control experiments. This strategy also helps to capture non-Markovian features, such as non-exponential dwell times, as discussed in our previous work on ribosome diffusion (<https://doi.org/10.1038/s41467-022-29515-x>).

In the context of this study, our microscopy data for IF2 α , IF2 γ , and IF3 (the main focus of this work), consistently reveal at least four distinct diffusion states: (1) free factors, with factor-dependent diffusion coefficients $>1 \mu\text{m}^2/\text{s}$; (2) a slower state, likely corresponding to "IF-30S" (factors bound to ribosomes not yet engaged with mRNA); (3) an even slower "IF-30S-mRNA" state (factors associated with ribosomes bound to transcripts); and (4) a very fast-diffusing population, similar to free HaloTag, likely representing proteolytic fragments. These states are robustly detected across independent experiments and different IFs in HMM models with 5 or more diffusion states (see Supplementary Data 8-10), supporting the robustness of this state classification.

In the revised manuscript, we now discuss this in detail (see sections "HMM analysis of single-molecule trajectories" and "Ribosome binding kinetics of IFs and fMet-tRNA^{fMet}") and have, for clarity, chosen to present all

data using 5-state HMM models, that are coarse-grained into 3 states: “factors associated with ribosomes”, “freely diffusing factors”, and “cleavage products”. Furthermore, we show that after coarse-graining into these 3 states, the exact model parameters may vary to some extent, but the overall conclusions remain consistent when coarse-graining is performed on HMM models with 5 or more diffusion states (see Supplementary Fig. 9, 12).

Although we do observe diffusion states which likely correspond to "IF-30S" and "IF-30S-mRNA", and their relative occupancy appear to be fairly consistent across different HMM models, our attempts to coarse-grain larger models into four diffusion states result in models with highly model-dependent kinetic parameters. That is, the transition rates between those states, and thus the estimated dwell times, are not robust (see new Supplementary Fig. 11), unlike in our previous study on tracking of ribosomes. A possible explanation is that the 5 ms camera exposure time used in this study enables the detection of short binding events and the tracking of smaller, faster-diffusing molecules, but is less effective at resolving transitions between slow-diffusion states. In contrast, longer exposure times used in our previous ribosome tracking study were better suited for capturing such transitions. Therefore, in this work, in order not to risk over-interpreting the results, we report IF-bound state dwell times as the average duration of binding to 30S subunits in both these sub-states. We note that binding of mRNA has been shown to occur independent of initiation factors and can take place at any time during formation of the initiation complex (<https://doi.org/10.1038/nsmb.2285>). Hence, we believe that our results remain relevant.

5. A definition of “immature” and “mature ribosomes” should be provided to help understand claims and interpretations and to clarify why the authors assume that immature ribosomes are tethered, and specifically via RNAP (and not via nascent RNA).

Response 2.5:

In the revised manuscript, we decided to avoid usage of the term “immature ribosome”, since it covers a wide range of ribosomal particles at various stages of their biogenesis, and might refer to different intermediates in the literature depending on the context or experimental system. Instead, we provided a more detailed description in the text of the specific stage of ribosome assembly we aimed to investigate: incomplete 30S subunits that are still associated with nascent RNA during the early stages of rRNA synthesis. Please see revised section “The N1 domain of IF2 binds to NusA”.

6. Often in the manuscript there is a mention for 30S binding, but it is unclear whether binding to the 30S occurs while the 30S is on or off the mRNA. This is important to establish, since if the binding occurs on a 30S ribosomal subunit that is not engaged with the RNA, there is no connection to the translation cycle since the step that may be affected precedes association with the nascent RNA. The authors should also comment on the possibility that IF2 α enters the RNA as a complex with another protein or even the 30S subunit.

Response 2.6:

We think that the reviewer raises an excellent point. In our previous study on ribosome tracking, we were indeed able to successfully distinguish between free ribosomal subunits and those engaged with mRNAs. Although, we do detect distinct diffusion states which likely correspond to "IF-30S" and "IF-30S-mRNA", the kinetic parameters for larger coarse-grained models which differentiate those states were not robust across different HMM model sizes. Please see **Response 2.4** for a more detailed explanation, as well as the extended section "Ribosome binding kinetics of IFs and fMet-tRNA^{fMet}", including new Supplementary Fig. 11.

*7. NusA is also a major antitermination factor, with high occupancy on rRNA operons. The results interpretation and discussion need to account for this. What prevents IF2 α from binding there or from binding to NusA in the cytoplasm? Further, the ms needs to discuss/cite in the putative formation of NusA condensates (see Ref. 57), the involvement of NusA as the putative bridging factor in TTC in *M. pneumoniae* cells (<https://pmc.ncbi.nlm.nih.gov/articles/PMC7115962/>), and the Mooney et al ChIP-chip work on NusA (<https://doi.org/10.1016/j.molcel.2008.12.021>).*

Response 2.7:

We agree with the reviewer that our discussion of the functional roles of NusA was rather limited and did not include several important earlier and more recent studies. We have now significantly expanded the discussion section to provide a more comprehensive overview of NusA, including its domain organization, involvement in antitermination complex, role in formation of condensates, as well as its involvement in different modes of TTC (see revised "Discussion"). Furthermore, we included additional experiments where we track a Δ NTD-NusA mutant, which helps us to understand if it can interact with the ribosome and with the IF2 α isoform in an RNAP-independent manner (see **Response 2.2** and section "The N1 domain of IF2 binds to NusA").

8. Pg 14: The results show that NusA interacts with the chromosome (which is expected) and that IF2 α interacts with the chromosome as well. This result is one of correlation/colocalization, and does not prove an interaction; this distinction should be clarified in the ms.

Response 2.8:

To address this point, we have revised the second paragraph of the section "The N1 domain of IF2 binds to NusA". We would also like to clarify that it was not our intention to imply that similar spatial distributions alone are sufficient to demonstrate molecular interactions.

9. A more detailed explanation for the rationale behind projecting only the middle 40% of dot coordinates is required – to what extent is the data affected by this choice? This is also important in the context of the RNAP

distribution in vivo, where it has been shown that transcribing polymerases are in the periphery of the nucleoid (<https://doi.org/10.1073/pnas.1507592112>).

Response 2.9:

We thank the reviewer for pointing at this issue. We revisited previous publications on super-resolution imaging of ribosomes and PNAP, which then motivated us to improve the spatial distribution plots. In particular, it has been shown by previous research, that locations and the number of nucleoids are different between cells in different stages of their growth but also between cells at similar stages of their division. We note that in our data, the spatial distributions were obtained from a large number (hundreds or thousands) of cells, which results in “blurring” of potentially valuable observations. Therefore, we decided to apply size filtering of cells in our analysis (only for spatial distributions) to discard those which have just divided (which might have a single nucleoid in the middle or 2 nucleoid lobes that are poorly separated in space), as well as larger cells (which often have > 2 nucleoid lobes, as shown in <https://doi.org/10.1111/j.1365-2958.2012.08081.x>). Furthermore, to address the point raised by the reviewer, we revised our presentation of the spatial distribution data and now provide heatmaps of dot locations using normalized cell coordinates. We hope that this format offers a more intuitive and visually accessible representation of the localization patterns and also doesn't involve any exclusion of data as it was in the original version of the manuscript.

10. The authors allude to the NusA-IF2 α mediated coupling contributing to a rewiring of the transcriptome and translome without any analysis of gene expression.

Response 2.10:

We have removed these sentences from the manuscript.

11. To improve clarity, the authors could extend the figures in the main text, e.g. incorporate Supplementary Figure 6 into Figure 3, to reduce redundancy and prevent flicking back-and-forth. Additionally, they could include the sample sizes (n values) into the main text, rather than in the supplementary material. A schematic of translation initiation would have been helpful in the intro (include in Fig 1), further, a schematic diagram of their proposed model may improve audience understanding.

Response 2.11:

To address this point, we have substantially modified the figures in the main text and in the Supplementary Materials. In the revised version of the manuscript, we significantly extended our explanation regarding the data analysis and therefore additional figures were provided (new Supplementary Fig. 2-6, 9, 11, 12). Furthermore, in Fig. 1 we provided a schematic overview of translation initiation (Fig. 1a), and included an additional panel Fig. 1c showing an example of HMM fits. Fig. 3 now includes all key spatial distribution plots from the original version of the manuscript. We think that the spatial distribution plots for the wild-type IF2 α and IF2 γ do not contribute

significantly to the main conclusions of our work and, thus, have remained in the supplementary. Spatial distribution plots for fast diffusion states were removed due to the minimal contribution and to avoid confusion. Fig. 4 now includes domain organization of NusA – a key factor of this study. Besides that, Fig. 4 now shows results of the new experiments discussed in **Response 2.1**. Figure legends were extended to refer to relevant Supplementary Data and provide sample sizes of microscopy data.

12. The article may benefit from expanding on the relevance of adaptation mechanisms to changing growth conditions to further the discussion and highlight the importance of their findings.

Response 2.12:

We have extended the discussion to further highlight the importance of fast adaptation to changing growth conditions. Specifically, we now discuss how nutrient fluctuations, such as carbon source shifts, trigger metabolic stress responses like the stringent response (see revised last paragraph of “Discussion” section).

Minor

1. The figures need improvement; the font sizes and some of the figure elements are too small.

Response 2.13:

As mentioned above, the figures and figure legends have been revised. In the process, we aimed to improve visual clarity by using larger fonts and following the general formatting guidelines of Nature Communications.

2. An example of the HMM fits for at least one factor needs to be in the main text to showcase the approach and its limitations.

Response 2.14:

We included an additional panel in Fig. 1 (Fig. 1c), to address this point.

3. The length of the trajectories needs to be discussed since it does provide an indication of the observation span and the timescales of transitions that can be observed.

Response 2.15:

We addressed this point in the section “HMM analysis of single-molecule trajectories” and also provided a Supplementary Fig. 3.

Reviewer #3 (Remarks to the Author):

Response 3.0:

We thank the reviewer for reading and reviewing our manuscript. We hope the revised version addresses all concerns and improves the clarity of our work

Reviewer #4 (Remarks to the Author):

The manuscript “A complex between IF2 and NusA links transcription and translation” by Metelev and Johansson presents a plethora of experiments leveraging cutting-edge microscopy technique to probe into the behaviour and interaction of the translation initiation factors in live E. coli cells. Particularly interesting is the newly found connection between the lack and altered functionality of specific IFs and the adaptation of E. coli to new growth conditions in the form of prolonged lag phases. This study convincingly shows how single-particle tracking (spt) can enrich the study of biological systems in situ. Overall, this is a very well written and technically strong manuscript in which the authors included many controls to back up their biological statements. In the following, I want to focus a bit on the technical aspects of the study, which I hope can be addressed by the authors.

Response 4.0:

We thank the reviewer for carefully reading and evaluating our manuscript, and for the encouraging feedback. In response, we have thoroughly revised the manuscript, including substantial updates to data presentation and analysis. Particularly, to address concerns about reproducibility, we now explicitly show results from independent experimental repeats. We have also expanded on explanations throughout the manuscript to improve clarity. We hope that these revisions fully address the reviewer’s concerns outlined below.

Major comments

- As mentioned, the authors performed many spt experiments in many different genetic backgrounds. If I understand correctly, rather than using conventionally calculated (apparent) diffusion coefficients to infer the mobility of complexes, the authors fed the localisations and tracks into a hidden Markov model (HMM) to obtain dwell times and diffusion coefficients. Whereas overall some of the obtained results are in line with previous reports, the current version of the manuscript makes it difficult to judge for the reader how well the occupancies obtained by HMM describe the raw data obtained through single-particle tracking. To this end, I would suggest generating more conventional diffusional distributions of their tracks and provide them as supplementary material. As such, a qualitative evaluation of the diffusional equilibrium can be used to gauge whether the relative occupancies in slow and fast states provided by the HMM fitting are actually a good interpretation. Along the same lines, and maybe the

authors have done that already, a comparison of the HMM output from the individual repeats (all measurements were at least measured in triplicates) could help to built further trust into the HMM data. As of now, the HMM, albeit well-described in literature, is a black box that will always provide output with the challenge being the correct interpretation.

Response 4.1:

We thank the reviewer for valuable feedback and constructive suggestions. As correctly summarized, we performed single-particle tracking experiments on various molecules, primarily those involved in translation initiation. The use of bright and photostable fluorophores allowed us to track molecules for significantly longer durations compared to traditionally used fluorescent proteins. Even a direct visual inspection of microscopy movies reveals that the particles do not diffuse homogeneously, but instead occupy distinct, long-lived diffusion states (see Movies S1-S6). Notably, we also observe transitions between these diffusion states within individual movies. Therefore, we wanted to take advantage of the Hidden Markov modeling (HMM) approach to capture not only diffusion states, characterized by different diffusion coefficient, but also to estimate the rates of transitions between these states. By quantifying the frequency of these transitions, we were able to calculate the average dwell times associated with each diffusion state. This makes the HMM approach superior to step-length analysis and mean-square displacement (MSD) analysis for this work. That said, we agree with the reviewer that comparing HMM with other methods can further strengthen our analysis and increase transparency. In the revised manuscript we have included step length distributions within diffusion trajectories of labeled factors (see Supplementary Fig. 2). We opted for step-length distributions over MSD analysis due to the likelihood that individual trajectories contain mixed diffusion states, and we observed considerable variability in MSD fitting results depending on how trajectories are fragmented, making step-length analysis a more consistent and interpretable comparison in our case.

Furthermore, to address the next point raised by the reviewer, we have revised our analysis and data presentation to include individually performed replicate experiments. We also evaluated the statistical significance between groups using a standard two-sided unpaired t-test and included those in the manuscript.

- Regarding the statistics, maybe rather than just talking of minicolonies that were imaged, could the authors state the number of cells imaged per condition and the number of tracks per condition?

Response 4.2:

We thank the reviewer for this suggestion. In the figure legends, we have included information about the number of cells containing tracked molecules per experimental condition, as well as the total number of trajectory steps.

- The use of HaloTag enables the generation of long tracks, reaching even second-long durations. The authors combined the HaloTag with fast exposures and frame times, being able to also resolve interactions as short as 20 ms within their tracks. At the same time, the HMM applied here to fit their single-particle tracking data seems to

account for a balance between slow and fast states of the imaged proteins at the steady-state. Does this approach exclude the possibility that the same protein changes of state while it is being visualised (within the same track), which could be the case for some conditions in this manuscript? If state transitions are indeed a common occurrence due to track length and average dwell times, how is their HMM approach resistant to this issue? I would further suggest including some example histograms of track lengths to get a better idea how long the average track can be.

Response 4.3:

We thank the reviewer for this important question! One of the main advantages of using HaloTag and other stable fluorophores is that, indeed, observed tracks contain transitions between different diffusion states which enables us to calculate transition probabilities between states and estimate dwell times in diffusion states. First, to address this question, we provide additional Supplementary Fig. 4 which shows that transitions between states are commonly observed within microscopy movies. Tracks shown in this figure are taken from Movies S2-S4 which are provided with the manuscript. Furthermore, we extended our explanations in the section “HMM analysis of single-molecule trajectories” to improve clarity. We use an algorithm in which all trajectories are fitted using global maximum-likelihood estimation, to a model with a pre-defined number of hidden diffusion states. Importantly, diffusion coefficients and transition frequencies between the states within trajectories are used as fitting parameters. We note though, that dwell times in different states are calculated based on the HMM fitted frequencies of transitions between different states, and not on duration of complete binding events. Hence, with the assumption that the system of study is in steady-state, it is possible to estimate longer binding events than the recorded trajectory lengths. Our manuscript cites the fundamental work regarding this HMM analysis (<https://doi.org/10.1038/ncomms15115>).

Minor comments.

*- In the introduction, the authors mention the different isoforms of IF2. It is however unclear to me whether they come from different genes or whether other processes are involved in the generation of IF2 α , IF2 β and IF2 γ from the same *infA* gene. Could the author add a brief explanation in the introduction section on how the isoforms are generated? In the abstract, the authors refer to two naturally occurring isoforms, so that is a bit confusing or am I missing something?*

Response 4.4:

We have now clarified in the text how the different isoforms of IF2 are produced from a single *infB* gene through the use of multiple alternative in-frame initiation codons.

The following sentence was added to the introduction: “In *E. coli*, the *infB* gene contains three alternative in-frame translation initiation codons, leading to the production of three IF2 isoforms. The two smaller isoforms, IF2 β (79.9 kDa) and IF2 γ (78.8 kDa), differ by only seven amino acids in the N-terminus, while the significantly larger IF2 α isoform (97.3 kDa) includes an additional 157-amino-acid-long Domain I.”

- Page 4. Fig.1 / Second paragraph results section and throughout. Please indicate wheter IF was tagged on the chromosome or was made available on a plasmid.

Response 4.5:

We have clarified this point in the revised manuscript. For tracking of IFs, we used plasmids with very low expression levels of HaloTag-labeled proteins.

- Page 11. In the sentence “Quite remarkably, a single mutation, S753Y, renders IF2 to preferentially adopt the active conformation independent of fMet-tRNA^{fMet} 24,45.” there seem to be a verb missing, the sentence does not make sense as is.

Response 4.6:

We replaced the word "renders" with "causes" for improved grammatical accuracy and clarity.

- Page 14. In the sentence “As an initial naïve test to find the additional binding partner of IF2a, we decided to...”, IF2a is supposed to be IF2alpha.

Response 4.7:

Corrected.

- Page 14. In the “The N1 domain of IF2 binds to NusA” Results section, the fusion of RbfA and HaloTag is first spelled as RbfA-HaloTag and then as HaloTag-RbfA. Often the spelling of the fusions indicates at what terminus the proteins are fused. Could the author provide a consistent naming, also consistent with the terminus at which the HaloTag is fused to?

Response 4.8:

Thank you! Corrected. It is RbfA-HaloTag.

- Page 15. “As mentioned previously,... previously” Delete one instance. Add reference here.

Response 4.9:

Corrected.

- Page 25. “model sizes 6-9” What does that mean?

Response 4.10:

In the revised manuscript, we now discuss in more details which diffusion states are identified by HMM (see sections “HMM analysis of single-molecule trajectories” and “Ribosome binding kinetics of IFs and fMet-tRNA^{fMet}”).

In the original version of the manuscript we avoided usage of a particular model size for showing the results. Instead we used weighted averages calculated from larger HMM models, including model sizes from 6 states to 9 states. Although, it is not clear what approach is better, in order to improve the clarity of the manuscript we significantly modified our analysis and how data are presented.

For clarity and simplicity, throughout the manuscript we now discuss coarse-grained results for occupancies and dwell times using a model size of 5 diffusion states, where for IFs we observe a robust separation into three clusters: ribosome-bound, free, and cleavage products (please see sections “HMM analysis of single-molecule trajectories”). Furthermore, we show that our conclusions are generally model-independent for different HMM model sizes (Supplementary Fig. 9, 12).

- In the sample preparation sections of Materials and Methods, cells are diluted to a final OD before being deposited on agarose pads to form microcolonies. In the section “Sample preparation for in vivo labeling using HaloTag” the final OD value is 0.003, whereas in the section “Cell electroporation with Cy5-labeled molecules” the final OD value is 0.03. Are the two final OD values actually

Response 4.11:

We thank the reviewer for careful reading. What is stated in the Materials and Methods is correct. The reason is that after electroporation ~90% of the cells die due to electroporation, those cells contribute to the OD measurement calculations, but they do not grow and divide to mini-colonies (and are also discarded in the analysis as they get stained with Sytox-Blue dye). In other words, we put approximately the same number of alive cells in both procedures.

NCOMMS-24-65154-A

Response to reviewers' comments

Dear Dr Johansson,

Thank you for submitting your manuscript "A complex between IF2 and NusA links transcription and translation" to Nature Communications. I am delighted to say that we are happy, in principle, to publish it under an open access license.

First, we ask you to revise your paper to address our editorial requests (in the attached Author Checklist) and any remaining comments from reviewers (included at the end of this email, if applicable).

Response:

We thank the editor and the reviewers for their constructive feedback throughout the process, which we think have improved the quality of the manuscript considerably. In addition to providing the technical details requested in the Author Checklist, we have addressed the remaining concerns from Reviewer #2 regarding the interpretation of the results. Please see details below.

REVIEWER COMMENTS

Reviewer #2 (Remarks to the Author):

The authors have taken on board the comments and made many changes in the text to clarify their findings; they also added new results and analysis. In particular, the authors have (for the most part) appropriately adjusted their conclusions to suggest a potential role of IF2 α -NusA in coupling transcription and translation rather than a direct linkage. However, there are still some instances where the language has not been corrected, e.g. the title remains unchanged, and see line 78: "Hence, our results suggest a hitherto unknown mechanism for transcription-translation coupling during translation initiation." Toning down these statements before publication will be appropriate.

Response:

We thank the reviewer for all the constructive criticism and suggestions. We have now changed the title to tone down the findings. Considering the rather strong evidence of a physical link between IF2 α and NusA, and the results

shown in Fig. 4h, suggesting that IF2 α but not IF2 γ participate in initiation at the chromosome, we, however, believe that it is still appropriate to speculate that our finding suggests early coupling of transcription-translation via this interaction. The sentence on line 78 has been modified and is now more in line with the title and the abstract.

Reviewer #3 (Remarks to the Author):

Response:

We thank the reviewer for all the help in improving the quality of the study.

Reviewer #4 (Remarks to the Author):

The authors addressed all my concerns and suggestions. By adding more explanation on the HMM data analysis, comparisons with conventional step sizes distributions, and more statistical rigour the manuscript now makes an(other) excellent case for using single-molecule resolved imaging to elucidate biological functions and pathways.

Response:

We thank the reviewer for all the help in improving the quality of the study.